# AOD data fusion with Geostationary Korea Multi-Purpose Satellite (GEO-KOMPSAT-2) instruments GEMS, AMI, and GOCI-II: Statistical and deep neural network methods

Minseok Kim[1], Jhooon Kim[1], Hyunkwang Lim[2], Seoyoung Lee[3,4], Yeseul Cho[1], Yun-Gon Lee[5], Sujung Go[3,4], Kyunghwa Lee[6]

[1]Department of Atmospheric Sciences, Yonsei University, Seoul, 03722, South Korea
[2]National Institute for Environmental Studies (NIES), Tsukuba, 305-0053, Japan
[3]Goddard Earth Sciences Technology and Research (GESTAR) II, University of Maryland Baltimore County, Baltimore, 21250, MD, USA
[4]NASA Goddard Space Flight Center (GSFC), Greenbelt, 20771, MD, USA
[5]Department of Atmospheric Sciences, Chungnam National University, Daejeon, 34134, South Korea
[6]National Institute of Environmental Research (NIER), Incheon, 22689, South Korea

*Correspondence to*: Jhoon Kim (jkim2@yonsei.ac.kr)

**Abstract.** Data fusion of aerosol optical depth (AOD) datasets from the second generation of Geostationary Korea Multi-Purpose Satellite (GEO-KOMPSAT-2, GK-2) series was undertaken using both statistical and deep neural network (DNN)-based methods. The GK-2 mission includes an Advanced Meteorological Imager (AMI) onboard GK-2A and a Geostationary Environment Monitoring Spectrometer (GEMS) and Geostationary Ocean Color Imager-II onboard GK-2B. The statistical fusion method, Maximum Likelihood Estimation (MLE), corrected the bias of each aerosol product by assuming a Gaussian error distribution, and accounted for pixel-level uncertainties by weighting the root-mean-square error of each AOD product for every pixel. A DNN-based fusion model was trained to target Aerosol Robotic Network AOD values using fully connected hidden layers. The MLE and DNN AOD outperformed individual GEMS and AMI AOD datasets in East Asia (R = 0.888; RMSE = −0.188; MBE = −0.076; 60.6% within EE for MLE AOD; R = 0.905; RMSE = 0.161; MBE = −0.060; 65.6% within EE for DNN AOD). The selection of AOD around Korean peninsula, which is incorporating all aerosol products including GOCI-II resulted in much better results (R = 0.911; RMSE = 0.113; MBE = −0.047; 73.3% within EE for MLE AOD; R = 0.912; RMSE = 0.102; MBE = −0.028; 78.2% within EE for DNN AOD). The DNN AOD effectively addressed the rapid increase in uncertainty at higher aerosol loadings. Overall, fusion AOD (particularly DNN AOD) showed improvements with less variance and a negative bias. Both fusion algorithms stabilized diurnal error variations and provided additional insights into hourly aerosol evolution. The application of aerosol fusion techniques to future geostationary satellite projects such as TEMPO, ABI, and GeoXO may facilitate the production of high-quality global aerosol data.

# 1 Introduction

Since the launch of the Advanced Very High-Resolution Radiometer (AVHRR) onboard the US National Oceanic and Atmospheric Administration (NOAA) satellite, various atmospheric aerosol remote sensing techniques have been developed using spaceborne sensors (Kaufman et al., 1990; King et al., 1999). Radiometers such as the AVHRR, the MODerate resolution Imaging Spectroradiometer (MODIS), and the Visible Infrared Imaging Radiometer Suite (VIIRS) observe spectral bands across the visible (VIS) to infrared (IR) range. Several algorithms have been developed for these instruments to quantify aerosol optical depth (AOD) on a global scale. The Dark Target (DT; Kaufman et al., 1997; Remer et al., 2005; Levy et al., 2013) and Deep Blue (DB; Sayer et al., 2013; Hsu et al., 2006, 2013) algorithms, designed for aerosol optical property retrieval from MODIS, have established a global standard for spaceborne AOD products. The Multi-Angle Implementation of Atmospheric Correction (MAIAC) conducts spatiotemporal combinations of observations to facilitate aerosol retrieval and atmospheric correction (Lyapustin et al., 2011a, b; 2012; 2018). These algorithms have also been adapted for use with the VIIRS, which has similar observation specifications to those of MODIS. The Multiangle Imaging Spectroradiometer (MISR) measures surface radiance from nine viewing angles, enabling aerosol optical property inversion, and providing much information about different aerosol types (Kahn et al., 2001). Previous studies have exploited the sensitivity of hyperspectral imaging capabilities in the ultraviolet (UV) to VIS range of instruments such as the Total Ozone Mapping Spectrometer (TOMS), the Ozone Monitoring Instrument (OMI), and the TROPOspheric Monitoring Instrument (TROPOMI) to detect absorbing aerosols such as smoke and dust (Torres et al., 1998, 2002, 2020; Ahn et al., 2014). These aerosol retrieval algorithms involved the UV Aerosol Index (UVAI) in identifying aerosol types and derive AOD using pre-computed reflectance for the selected aerosol type; the algorithms employed by these instruments provide AOD as well as information on single scattering albedo (SSA), aerosol layer height (ALH), and above-cloud AOD (Torres et al., 2012; Jethva et al., 2018). The Deep Space Climate Observatory (DSCOVR) is located at the Lagrange-1 point, allowing continuous observation of Earth's sunlit area. The Earth Polychromatic Imaging Camera (EPIC) onboard DSCOVR features 10 bands spanning 317–779 nm. Lyapustin et al. (2021) demonstrated the capability of MAIAC in retrieving AOD and SSA from such instruments and further quantified the content of iron oxide in atmospheric mineral dust (Go et al., 2022).

For the Geostationary Earth Orbit (GEO) observations in Asia, the GEO-Kompsat-1 (GK-1) satellite, also known as the Communication, Ocean, and Meteorological Satellite (COMS) was launched in June 2010, equipped with a Meteorological Instrument (MI) and a GOCI (Kim et al., 2007; Lee et al., 2010b). The first satellite of the second series, GK-2A, was launched in December 2018, featuring an Advanced Meteorological Imager (AMI; Kim, D. et al., 2021). GK-2B was launched in February 2020, carrying the successor to GOCI (GOCI-II) and a Geostationary Environment Monitoring Spectrometer (GEMS) (GOCI-II ref; Kim et al., 2020; Choi et al., 2021). As the retrieval skill of the aerosol algorithm for GK-1 has well been established (Kim et al., 2008; Lee et al., 2010b; Kim et al., 2016; Choi et al., 2016, 2018), the instruments onboard GK-2 continue aerosol monitoring with improved observation specifications. For GEMS, as the first

geostationary hyperspectral spectrometer, an aerosol algorithm based on a two-channel inversion with an optimal estimation approach has been developed (Kim et al., 2018).

Aerosol retrieval algorithms vary in their application with the spectral specifications of the sensors they use. Employing multiple channels within the VIS to near-infrared (NIR) range enables aerosol inversion to capture information on aerosol size (Lin et al., 2021), whereas UV–VIS spectral observations are sensitive to aerosol absorption and layer height (Kim et al., 2018). Variability of instrument sensitivity leads to distinct error characteristics in aerosol algorithms. For example, retrieval algorithms utilizing wavelengths less sensitive to ALH tend to be robust against assumptions about ALH during radiative transfer simulations. Moreover, observations across different wavelength ranges influence the spatial coverage of retrievals. As aerosol loading increases over a dark surface, atmospheric backscattering intensifies, causing more photons to reach satellite instruments. Conversely, over a bright surface, an elevated aerosol loading obscures signals reflected from the surface. The balance, where the increase in backscatter and disturbance of surface signals align, is termed the critical reflectance, which decreases with increasing wavelength (von Hoyningen-Huene, et al. 2011; Kim et al. 2014). As most land surfaces have lower reflectance at shorter wavelengths, aerosol retrieval at such wavelengths results in broader spatial coverage.

Previous studies have shown that the use of multiple aerosol products addresses a systematic error tendency in AOD retrieval. MLE merging of AOD data from two or more satellites (or algorithms) has been explored in enhancing the accuracy of AOD quantification. Levy et al. (2013) and Wei et al. (2019) produced a merged MODIS DT–DB AOD product, accounting for systematic biases from each algorithm. Tang et al. (2016) employed the Bayesian maximum entropy method to merge AOD from MODIS and Sea-viewing Wide Field-of-view Sensor (SeaWiFS) while considering the retrieval uncertainties associated with each AOD product. The optimal interpolation method iteratively updates AOD, factoring in MODIS, SeaWiFS, and MISR AOD uncertainties (Xue et al., 2014). Gupta et al. (2008) considered the point-spread function of each satellite footprint as a weighting factor for the merging of AOD from MODIS, MISR, and Clouds and the Earth's Radiant Energy System (CERES). The Maximum Likelihood Estimation (MLE) technique, which maximizes a cost function defined by the Gaussian error distribution of satellite AOD products, is widely used in merging AOD data. This method was applied by Xu et al. (2015) to MODIS, SeaWiFS, and MISR data, and by Go et al. (2020) to OMI and MODIS data, accounting for uncertainties in each pixel. Xie et al. (2018) improved the methodology by correcting the systematic biases of the Advanced Along-Track Scanning Radiometer (AATSR). aerosol algorithms. Lim et al. (2021) employed the MLE method in considering pixel-level uncertainty and bias correction, resulting in merged AOD products from Advanced Himawari Imager (AHI), Geostationary Ocean Color Imager (GOCI), and TROPOMI systems.

Most satellite AOD data-fusion research has concentrated on Low Earth Orbit (LEO) satellite data with limited consideration of GEO data. Unlike the LEO satellites, hourly variations in retrieval uncertainty emerge when using continuous AOD data from GEO satellites, and accounting for diurnal variations in uncertainty in aerosol data fusion is challenging. Furthermore, error characteristics among different AOD products under different retrieval conditions complicate matters. Deep learning excels in capturing nonlinearities owing to its hierarchical architecture and activation functions in each layer, so a deep

neural network (DNN) approach may significantly enhance AOD fusion outcomes. However, exploration of deep learning approaches to AOD data fusion has been limited. In this study, both a conventional MLE AOD fusion algorithm and a DNN-based AOD fusion algorithm have been developed and validated using aerosol products from GK-2 satellites. Due to differences in spatial domain of each instrument, fused datasets are validated separately in a region around the Korean peninsula (KO) and a region within the East Asia (EA). Section 2 briefly introduces the spaceborne AOD datasets used in this study, derived from aerosol retrieval algorithms and an AERONET AOD dataset. Each fusion method is described in Section 3, and Section 4 discusses the fused AOD products based on diagnostic and prognostic error analysis. Finally, Section 5 provides a summary of the overall results and outlines prospects for geostationary aerosol data fusion.

## 2 Data

### 2.1 GK-2 Satellite instruments

An overview of instruments onboard GK-2 satellites and their aerosol products is provided in Table 1, and Fig. 1 illustrates the data coverage of each aerosol product.

#### 2.1.1 AMI/GK-2A

As a meteorological imager, AMI has spectral channels in the VIS–IR range (Kim et al., 2021), which is 3 VIS channels, 1 near-IR channel, and 10 IR channels from 0.47 $\mu$m to 13.3 $\mu$m. A 0.65 $\mu$m channel has 0.5 km spatial resolution, and 0.47, 0.51 $\mu$m channels has 1.0 km spatial resolution. The IR channels has 2.0 km spatial resolution. AMI scans full-disk every 10 minutes, and local area near Korean peninsula every 2 minutes. Similar to conventional aerosol algorithms for instruments with VIS–IR capabilities, such as MODIS, VIIRS, AHI, and Advanced Baseline Imager (ABI), the AMI aerosol algorithm employs the VIS and NIR channel for aerosol inversion while utilizing other channels for bright surface masking and surface reflectance estimation.

#### 2.1.2 GOCI-II/GK-2B

GOCI-II, onboard the GK-2B, is a second-generation instrument of GOCI. Compared to GOCI, GOCI-II features a better ground sampling distance, an extended field of regard covering the hemisphere, and more spectral bands covering wavelength from 380 to 865 nm. GOCI-II has 4 additional bands (380, 520, 620, and 709 nm) compared to GOCI. Moreover, a wideband channel of GOCI-II is used for star imaging to improve image navigation and registration quality. Image acquisition of GOCI-II around the Korean peninsula (2500 km × 2500 km) is done by scanning 12 slots 10 times per day.

### 2.1.3 GEMS/GK-2B

GEMS is the first UV-VIS hyperspectral instrument on GEO orbit. GEMS is designed to monitor air quality in Asia (5°S–45°N, 75–145°E). GEMS observes reflected hyperspectral solar radiance from 300 to 500 nm wavelength range with a
spectral resolution of 0.6 nm and a spatial resolution of 3.5 km × 7.7 km. GEMS scans sunlit part of the Earth during daytime. To obtain qualitative radiance data considering solar zenith angle, GEMS has 4 scanning scenarios of half-east, half-Korea, full-center, and full-west (Fig. 1). Since aerosol retrieval quality at high solar zenith angle depreciates, the half-east scan data, which is performed in the early morning, is not used for aerosol fusion study.

## 2.2 Aerosol retrieval algorithms for GK-2 instruments.

### 2.2.1 Yonsei aerosol retrieval algorithm (AMI, GOCI-II)

Lim et al. (2018) introduced the AHI Yonsei Aerosol Retrieval (YAER) algorithm, which was initially devised for ocean-color imagers. In this study, the AMI aerosol product was retrieved using the AHI YAER algorithm with minor modifications (Kim et al., 2024). The AMI YAER algorithm has two AOD retrieval versions based on different surface
reflectance estimation methods; i.e., the Minimum Reflectance Method (MRM) and the Estimated Surface Reflectance (ESR) method. The distinct advantages of AMI for aerosol retrieval include more accurate cloud and bright surface masking via its IR channels compared with other instruments. Furthermore, its Short-Wave Infrared (SWIR) observation capabilities enable the use of the ESR method for surface estimation, offering uncertainty characteristics that are distinct from those of the MRM. Ocean surface reflectance estimation uses the Cox and Munk method (Cox and Munk, 1954) to estimate water
leaving radiance with consideration of wind speed and chlorophyll-a concentration. In addition, the AMI has high temporal resolution, with full-disk scans every 10 min. For the purpose of aerosol data fusion, a domain spanning 10°S–50°N and 70°E–150°E of the AMI full-disk scan was chosen to encompass the GEMS field of regard.

GOCI-II aerosol product is derived from the GOCI-II YAER algorithm (Lee et al., 2023) based on the GOCI YAER algorithm introduced by Lee et al. (2010b) and improved by Choi et al. (2016, 2018). The enhanced spatial resolution of
GOCI-II compared to GOCI allows its aerosol products to be retrieved at a resolution of 2.5 km, capturing higher-resolution spatial aerosol features around the Korean peninsula. However, the smaller field of regard of GOCI-II restricts AOD fusion to within the GOCI-II domain when using aerosol products from all three instruments. Consequently, fusion AOD utilizing all three aerosol products was evaluated separately (Section 4). Due to the absence of SWIR channels, the GOCI-II YAER algorithm estimates surface reflectance using only the MRM, with the concentration of 12 spectral bands within the UV–NIR
range contributing to the stability of AOD inversion. Ocean surface reflectance estimation is done as the same manner for AMI YAER algorithm using the Cox and Munk method.

Aerosol size influences the spectral dependency of AOD within the UV-NIR spectral range. Therefore, both AMI and GOCI-II are sensitive to the potential misclassification of aerosol types related to their size. However, their reduced

sensitivity to aerosol absorption in the VIS range renders the YAER AOD products robust against uncertainties arising from aerosol absorptivity. Both the AMI and GOCI-II YAER algorithms consider four types of aerosols: black carbon (BC), non-absorbing (NA), mixed (MX), and dust (DU) aerosol. Aerosol models utilized in the YAER algorithms were derived from climatology of AERONET inversion dataset with the classification developed by Lee et al. (2010a). The YAER algorithm first retrieves AODs at all wavelengths within UV-NIR range and converted to 550 nm for all aerosol types. Then, aerosol type that shows minimum variance at 550 nm are selected aerosol type for the corresponding inversion pixel.

### 2.2.2 GEMS Aerosol retrieval algorithm

The GEMS aerosol algorithm was initially developed by Kim et al. (2018) and Go et al. (2020) based on synthetic data from OMI observations. The operational GEMS aerosol algorithm, based on real observations, was subsequently established by Cho et al. (2023). GEMS performs hourly hyperspectral radiance observations of 300–500 nm with 0.6 nm spectral resolution during daytime. Its spatial resolution at nadir point is 3 km × 7.7 km. Distinct from other instruments on GK-2 satellites, GEMS features near-UV measurement that can be utilized for aerosol inversion. The optically darker nature of desert surfaces in near-UV measurement serves as a favorable condition for the retrieval of aerosol signals from observed radiance. This allows unprecedented hourly aerosol monitoring over desert regions such as the Gobi and Taklamakan deserts. Furthermore, near-UV spectral region is known to be sensitive to aerosol absorption, contributing to distinct error characteristics in GEMS AOD relative to AMI and GOCI-II. In this study, a version 2 of the GEMS AOD at 550 nm was used to maintain consistency with AMI and GOCI-II AOD products. A version update of the GEMS aerosol algorithm was made public in November 2022, and earlier data were reprocessed accordingly.

Unlike the AMI and GOCI-II YAER algorithms, the GEMS is more sensitive to misclassification of absorbing/scattering aerosol types. The GEMS aerosol retrieval algorithm initially performs aerosol type selection with UV aerosol index (UVAI) and VIS aerosol index (VISAI). The algorithm assigns NA type to pixels with low UVAI values. The other pixels are separated into highly absorbing fine (HAF) type and DU type according to the VISAI values. The aerosol type classification of the GEMS AOD retrieval is superior to the other algorithms using visible wavelengths because of the sensitivity of UV wavelength to scattering characteristic of aerosols. Yet, relatively short range of observation wavelength in VIS region of GEMS compared to AMI and GOCI-II lacks sensitivity to aerosol size information. After the aerosol type classification, the algorithm performs a two-channel inversion used in OMI near-UV aerosol algorithm to derive first guess of AOD and SSA. Then, the first guesses are fed into the GEMS optimal estimation algorithm to retrieve AOD at 443 nm. The 443 nm AOD is converted to 550 nm AOD based on the selected aerosol type.

### 2.3 AERONET

The AErosol RObotic NETwork (AERONET) constitutes a global network of ground-based aerosol remote sensing instruments, with numerous sun photometer stations operating at various locations worldwide. The AERONET level 1.0 data

are unscreened measurement data. The cloud and pointing error screening is applied to level 1.0 data to produce a level 1.5 dataset. The level 1.5 data series are raised to level 2.0 (quality-assured) series after final calibration values are applied and manual data inspection is completed. Here, AERONET version 3, level 2.0 AOD products served as the target AOD for both MLE and DNN-based methods (Section 3). However, in the validation of individual and fused AODs (Section 4), AERONET version 3, level 1.5 AODs were used owing to the limited application of pre- and post-calibration for only a few sites up to 2023; therefore, level 1.5 AODs were more suitable for validating relatively recent data. The estimated uncertainty in precision in AERONET AODs is known to be 0.010–0.021 depending on the wavelength (Holben et al., 1998; Eck et al., 1999; Giles et al., 2019; Sinyuk et al., 2020). Over the course of the error analysis and training period, 74 stations reported ground-based level 2.0 AOD data. For the validation period, 91 stations reported ground-based level 1.5 AOD, including 28 stations within the field of regard of GOCI-II. Table 2 lists the AERONET sites used in the study. To match AOD wavelengths with those of spaceborne AOD products, the AERONET 550 nm AOD was derived through quadratic interpolation from AODs measured at 340, 380, 440, 500, 675, 870, and 1020 nm. For spatiotemporal matching with AERONET measurements, satellite data within a 25 km radius of each AERONET site were averaged, and AERONET AODs within 30 min of each exact hour were also averaged for spatiotemporal collocation (Park et al., 2020).

## 3 Methodology

A flowchart of MLE fusion and DNN-based fusion processes is shown in Fig. 2. Each of these fusion methods requires a pre-calculation process involving bias and uncertainty calculations for MLE fusion and a model training process for DNN fusion. Data spanning one year (November 2021 to October 2022) was used in pre-calculation processes. The resultant fused AODs were generated and validated for the period November 2022 to April 2023. Throughout data pre-calculation, AERONET AOD served as the reference ground truth for both fusion methods. The AMI Normalized Difference Vegetation Index (NDVI) was used as both an uncertainty source and an input for both fusion approaches, as calculated using Eq. (1):

$$NDVI = \frac{R_{red} + R_{NIR}}{R_{red} - R_{NIR}} \tag{1}$$

where $R$ represents AMI MRM surface reflectance. The MRM surface reflectance was used because aerosols may affect the NDVI when using observed reflectance. Spatiotemporal matching of AMI NDVI followed the same approach as AMI AOD. Henceforth, for the sake of simplicity, the statistically fused AOD is referred to as MLE AOD, and the DNN-based fused AOD as DNN AOD.

### 3.1 Spatiotemporal matching and additional cloud masking with AMI IR observations

Because each instruments observes the Earth radiance with distinct geolocation fields, the geolocations of aerosol products are different. Therefore, aerosol products were re-gridded into 0.05° × 0.05° grids by averaging AOD values from the three closest pixels located within a 0.15° radius of the center of each grid point. The choice of a 0.15° radius was intended to prevent grid pixels from becoming empty owing to the coarsening of spatial resolution near the scan edge (as in western

China). However, this approach may lead to smoothing of aerosol features in regions distant from scan edges. To counteract excessive smoothing and preserve small aerosol features after re-gridding, the strategy involved averaging the maximum three points regardless of how many points lay within the specified averaging radius. This spatial matching technique was intended to provide a balance, mitigating excessive smoothing while retaining finer aerosol features. The re-gridded aerosol products are also used for error analysis and DNN model training to account for small errors may induced from the re-gridding process.

Given the distinct scanning scenarios of each sensor, a distinct temporal matching strategy was employed for each AOD product to generate AODs for every precise hour between 00:00 UTC and 07:00 UTC. For temporal matching of 04:00 UTC fusion, GEMS AOD data scanned from 03:45 UTC to 04:15 UTC were utilized as the AOD representation for 04:00. In the case of AMI AOD, data were collected for a time span of 03:30−04:30 UTC from each precise hour and a median AOD was calculated. As for GOCI-II, data scanned at 03:15 and 04:15 were simply averaged.

Aerosol retrieval algorithms inherently include a cloud masking process; however, GEMS and GOCI-II cloud masking may exhibit errors owing to challenges in distinguishing thin clouds such as cirrus from aerosols using only VIS channels. Therefore, a cloud detection database employing IR channels was extracted during the AMI YAER algorithm retrieval process and applied to the GEMS and GOCI-II aerosol products. The cloud-masking criteria of the AMI YAER algorithm are shown in Table 3, where the first two criteria in the list utilize the fixed geometry of GEO satellites. Because clouds change rapidly with time, the maximum brightness temperature (BT) within the previous 10 days served as an estimate of BT on a clear day. Pixels displaying a difference between maximum BT during the 10 days and observed BT in the 6.9 and 11.2 μm channels of less than −28 K were thus identified as cloud pixels. This method was introduced by Kim et al. (2014) using MI and has proved effective and reliable with AHI (Lim et al., 2018). The 1.38 μm channel is highly sensitive to cirrus clouds (Roskovensky and Liou, 2003), so pixels exhibiting a top-of-atmosphere reflectance exceeding 0.35 in the 1.38 μm channel were masked as clouds. Detection of lower clouds involved the brightness-temperature difference (BTD) of the 13.3 and 10.3 μm (known as the "atmospheric window") bands (BTD10.3–13.3). Over clear pixels, the BT of the 13.3 μm channel is significantly lower than that of the 10.3 μm channel due to well-mixed $CO_2$ in the troposphere, resulting in a substantial BTD10.3–13.3. The presence of clouds reduces BTD10.3–13.3, as the BT of the 10.3 μm channel is lower in cloud pixels. The detection of higher clouds followed a similar approach utilizing the 12.3 μm channel, which is sensitive to high-altitude water droplets and ice crystals. The IR-based masks applied to GOCI-II and GEMS AODs were implemented across all aspects of the study including error analysis, bias and uncertainty calculations, and DNN model training.

The effects of a cloud mask in refining GEMS and GOCI-II AOD are shown in Fig. 3, where yellow boxes indicate cloud-free regions that were not removed by the additional cloud mask, and magenta boxes highlight regions where the original GEMS or GOCI-II aerosol algorithms inaccurately detected clouds, leading to overestimated AOD values. An example for 25 November 2022, over the arid region of northern China, is depicted in Fig. 3a–c; the Taklamakan desert is highlighted by the yellow box in Fig. 3a. In comparing the original GEMS AOD with that after application of the AMI IR cloud mask (Fig. 3b–c), it is evident that the cloud mask did not mistakenly classify bright surfaces as clouds. The magenta box in Fig. 3b

indicates areas where the GEMS aerosol algorithm retrieved AOD values over thin clouds, leading to significantly elevated values of up to 1.2, while values near the clouds remained below 0.2. Some pixels even had AOD values exceeding 2.0. These problematic pixels were removed in Fig. 3c, leading to spatially consistent AOD results for GEMS after application of the additional cloud mask. The additional cloud mask was applied to GOCI-II AOD on 9 March 2023 (Fig. 3d–f), when a substantial aerosol plume was being transported across the Southern Ocean toward the Korean peninsula. The hazy atmosphere extended over Japan and into the western Pacific Ocean. However, the GOCI-II YAER algorithm failed to accurately detect thin clouds (magenta box, Fig. 3e). Application of the AMI IR cloud mask (Fig. 3f) effectively removed cloud-contaminated AOD values. The yellow box (Fig. 3d) highlights a dense aerosol plume. GOCI-II AOD values over the plume remained intact after application of the additional cloud mask, demonstrating that the cloud mask based on IR channels was proficient in distinguishing thin clouds from thick aerosol plumes.

## 3.2 Statistical aerosol fusion: MLE AOD

MLE aerosol data fusion employed an MLE method that accounted for the pixel-level uncertainty of each aerosol product. The MLE method operates under the assumption that its input AODs have unbiased random errors. Typical AOD distributions, which are often lognormal, tend to have a Gaussian uncertainty distribution (Sayer et al., 2020). However, the actual mean error does not always coincide with zero, contradicting the assumption made by the MLE method. To enhance the MLE input data quality, a preliminary bias correction for each AOD product was undertaken before initiating the fusion process. Here, AOD bias was defined as the mean of a Gaussian distribution fitted to the AOD error, as compared with the collocated AERONET AOD. To account for bias characteristics attributed to optical path variations and surface conditions, AOD bias values were computed for each hour, aerosol type, and NDVI bin.

Based on a zero-mean Gaussian error assumption after bias correction, a log-likelihood function $\rho(\tau)$ was written as follows:

$$\rho(\tau) = \sum_i \frac{1}{R_i \sqrt{2\pi}} \left( -0.5 \left( \frac{\tau - \tau_i}{R_i} \right)^2 \right), \tag{2}$$

where $\tau_i$ is a bias corrected AOD from instrument $i$, and $R_i$ is the uncertainty of $\tau_i$. Then, a derivative of the above log-likelihood function was written as follows:

$$\frac{\partial \rho(\tau)}{\partial \tau} = \sum_i \frac{\tau - \tau_i}{R_i^2} . \tag{3}$$

Finally, the AOD that maximized the log-likelihood function had a $\tau$ value that made the above derivative zero:

$$\tau = \frac{\sum \tau_i R_i^{-2}}{\sum R_i^{-2}} \tag{4}$$

The above Eq. (4) can be interpreted as an uncertainty-weighted mean of AOD products. Here, the uncertainty associated with each aerosol product was represented by the root-mean-square error (RMSE) of the AOD products relative to

AERONET AOD measurements. As shown in Fig. 5-7, retrieval error does not increase (or decrease) linearly. Therefore, merging AOD datasets using the same RMSE value for all pixels is not desirable. The MLE fusion method linearizes the error characteristics by categorizing potential error sources such as AOD values, aerosol types, NDVI values, and observation times. The potential error source variables are selected based on the following logistics. First, AOD value itself and aerosol type is selected because as aerosol loading increases, aerosol model assumption affects retrieval performance.

Complex aerosol mixture at high aerosol loading leads to high uncertainty and aerosol retrieval algorithms have distinct aerosol model assumptions. NDVI is selected as possible error source to represent surface condition. Different surface types have different surface reflectance and surface types differentiate by vegetation amount and types (Hsu et al., 2013). Observation time difference in GEO measurements leads to distinct optical path of observed radiance. Therefore, GEO satellite AOD products have diurnal error variations (Lim et al., 2018; Zhang et al., 2020; Fu et al., 2023; Cho et al., 2024).

To deal with the uncertainty from this, observation time is selected as the possible error source. Based on this analysis, the bias of each AOD product was subtracted according to the NDVI value, selected aerosol type (ancillary output of each aerosol products), and observation time. Following this bias correction, the RMSE for the MLE procedure was computed. The output of the fusion process was then categorized in accordance with AOD data availability, as shown in Table 4.

### 3.3 Deep neural network-based aerosol fusion: DNN AOD

A DNN is a powerful tool for capturing non-linear relationships among physical variables. Although ground-based and spaceborne AODs exhibit linear relationships owing to their fundamentally similar physical meanings, their error characteristics under diverse retrieval conditions can introduce nonlinearity. The MLE AOD fusion method attempts to address this nonlinearity by considering pixel-level uncertainty associated with each aerosol product. However, certain unexplained nonlinearities remain, and a DNN-based AOD fusion algorithm was formulated as follows.

The DNN model was constructed to predict AERONET AOD as the target variable, employing the same input data as the MLE AOD fusion approach. To improve model convergence and enhance the overall performance of the DNN, a preprocessing step was necessary for the input data. This involved standardization of the NDVI, hour, and aerosol type index (for GEMS, 1 = HAF, 2 = DU, 3 = NA; for AMI and GOCI-II, 1 = BC, 2 = NA, 3 = MIX, 4 = DU). The standardization process was implemented using Eq. (5):

$$x_{input} = \frac{x - \mu_x}{\sigma_x},\qquad\qquad\qquad\qquad\qquad\qquad(5)$$

where $\mu_x$ and $\sigma_x$ represent the mean and standard deviation of input data $x$, respectively. The AOD follows a lognormal distribution skewed toward higher values. To address this distribution characteristic, a Box–Cox transformation was implemented for standardizing AOD products derived from the three instruments and AERONET. This transformation, based on the concept initially introduced by Tukey (1957), has been adapted and shown to be effective for data

normalization (Box and Cox, 1982; Sakia, 1992).

A simplified architecture of a fully connected feed-forward neural network model is illustrated in Fig. 2. This DNN model comprises three hidden layers, with each being fully connected. Within each hidden layer, batch normalization was implemented to avoid overfitting by bringing numerical data onto a common scale. In addition, the rectified linear unit (ReLU) served as the activation function. Weighting coefficients of the neural network were optimized by minimizing the mean-square-error (MSE) loss. During training, the backpropagation technique was applied, adjusting the weight coefficients based on the gradient of the loss function. Hyperparameters including batch size, number of neurons, and learning rate were determined using the asynchronous successive halving algorithm (ASHA; Li et al., 2020). For the ASHA optimization process, a maximum of 1000 trials were set, with a minimum of 100 trials. In each trial, half of the configurations were eliminated. Following optimization, the DNN model was trained for each case of AOD availability, as outlined in Table 4.

## 4 Results and discussion

### 4.1 Error analysis of GEMS, AMI, and GOCI-II AOD products.

Error analysis of original AOD products gives intuition of expected contributions of each AOD products and helps to interpret the outcome. Here, we used spatiotemporally matched AOD products to minimize the effect of re-gridding and temporal matching to the input AOD value. Wavelength of AOD used for error analysis and data fusion are at 550 nm. Also, additional IR cloud masking is applied to GEMS and GOCI-II AOD products. Fig. 4 depicts 2-dimensional histograms illustrating the match between individual AOD products and AERONET AOD measurements. The expected error envelope (EE envelope; $\pm(0.05 + 0.15AOD)$) of AOD was established by Levy et al. (2013). GEMS AODs exhibited a tendency to underestimate AOD at high aerosol loadings (Fig. 4a), with a slope of 0.429 relative to AERONET AOD. Cho et al. (2023) reported that the latest version of GEMS AOD at 443 nm does not have such a low slope, implying that the underestimation of GEMS 550 nm AOD in the version 2 algorithm may be due to either an algorithm issue or errors during wavelength conversion. Despite this underestimation, GEMS AODs were strongly correlated with AERONET AODs, with a Pearson's correlation coefficient (R) of 0.715. GOCI-II AODs yielded the most comparable outcomes to AERONET AODs among the four aerosol products (Fig. 4b). The stable inversion of AOD achieved through utilizing 12 UV–NIR channels likely contributed to the robustness of the GOCI-II YAER algorithm. The Mean Bias Error (MBE) of GOCI-II AODs was negative, mainly because of a clustering of slight underestimations at low aerosol loadings. This underestimation at low AOD is a known issue when using MRM surface reflectance because of the assumption that at least one aerosol-free day exists within a 30-day period, which is not universally valid because of background AOD as indicated in Lee et al. (2023). A comparable issue with low aerosol loadings was evident with AMI–MRM AOD (Fig. 4c). Both AMI–MRM and AMI–ESR AODs (Fig. 4c–d) displayed scattered patterns relative to GOCI-II AODs. The overestimation of low AOD values observed in both AMI–MRM and AMI–ESR AODs may be attributed to insufficient cloud masking over land. Comparison of the two AMI AOD products indicated that AMI–ESR AOD yields slightly superior outcomes, likely because of the enhanced surface

reflectance estimation over urban regions with the ESR method, as indicated in previous studies (Lim et al., 2018; Kim, M. et al., 2021).

The biases in GEMS AOD products with AOD, NDVI, aerosol type, and observation time are illustrated in Fig. 5, where blue numbers in each plot indicate the count of collocated data in the respective box–whisker, and green dashed lines in each panel correspond to the y-axis range of the corresponding panels in the GOCI-II error analysis (Fig. 6). As AERONET AOD values increased, the GEMS AOD acquired an increasingly negative bias. Conversely, at low aerosol loadings (AERONET AOD < 0.2), GEMS AOD displayed a positive skewness, implying that it tends to overestimate low AOD values while simultaneously overestimating high AOD values. Where NDVI < 0.5, the error in GEMS AOD consistently demonstrated a negative skewness and bias. However, in high-NDVI regions, usually associated with dark surfaces, the bias is nearer zero. The negative error of GEMS AOD for HAF aerosols may be induced by errors in aerosol optical properties of the model (Cho et al., 2023). However, aerosol type selection is not absolutely independent of surface conditions. In winter, the NDVI in Southeast Asia falls to 0.3–0.4 (Ji et al., 2017), with massive HAF aerosols being emitted by biomass burning (Yin, 2020). Then, GEMS AOD displays an M-shaped diurnal variation that is consistently negatively biased, except for at 01 UTC. Diurnal variations in GEMS AOD may be influenced by the relatively short atmospheric path length at noon (04 UTC). Furthermore, variations among collocated AERONET sites, which are due to differing scan scenarios throughout the day, may also contribute to the observed diurnal error variations.

The same error analysis applied to GOCI-II AOD is illustrated in Fig. 6. The GOCI-II AOD error relative to AERONET AOD displayed a pattern of underestimation with increasing aerosol loading (Fig. 6a), although the magnitude of this error was notably smaller than that observed with GEMS AOD. In terms of NDVI (Fig. 6b), GOCI-II AOD seemed to exhibit consistent behavior regardless of land surface conditions. Over ocean areas (NDVI <0) the GOCI-II YAER algorithm delivered unbiased retrievals, and bias characteristics were similar across different aerosol types. The MX-type aerosol (the most frequently selected type in the GOCI-II YAER algorithm) yielded the most stable results with the shortest range of whiskers (Fig. 6b). Conversely, the NA aerosol had the greatest whisker range, indicating potential issues with the NA model in the algorithm. Asian dust is associated with high aerosol loadings, and results for DU aerosols are slightly more negative (Fig. 6c). Diurnal variations in GOCI-II error appeared stable, with slight underestimation during mornings and late afternoons. As the GOCI-II field of regard is smaller than those of GEMS and AMI, geometrical conditions may have had less impact on its performance.

Error analysis results for the two versions of AMI AODs are depicted in Fig. 7, which shows that both AMI–MRM (Fig. 7a) and AMI–ESR (Fig. 7e) AOD biases tend to decrease with increasing AERONET AOD values. This bias pattern may be attributed to the difference in surface reflectance estimation methods used by the two AOD versions. This distinction became more evident with low aerosol loadings, where the surface signal contributes substantially to the observed radiance. A comparison of the initial box–whisker plots for each AMI AOD version suggested that the AMI–ESR AOD bias is closer to zero with low aerosol loadings. Furthermore, the shorter lengths of the box–whisker plots across various NDVI values (Fig. 7f) indicate that the AMI–ESR YAER algorithm provided a more consistent estimate of surface reflectance than that of the

AMI–MRM YAER. Considering that both versions of the AMI YAER algorithm employ the same aerosol models, the variations in AOD bias between the two were similar (Fig. 7c, g). The diurnal error variation (Fig. 7d, h) was not notably different between the two AMI AOD products. This similarity in diurnal error variations suggests that the choice of surface-reflectance estimation method has limited impact on error characteristics based on observation time.

## 4.2 Fusion data evaluation

### 4.2.1 Validation of the fused AOD with AERONET

Based on the error analysis and DNN model training for the period from November 2021 to October 2022, AOD data were fused for six months spanning November 2022 to April 2023. The GOCI-II field of regard focusing on KO was smaller than those covering EA, so the fused AOD utilizing GOCI-II AOD was confined within the domain. Therefore, two groups of fused AOD products were generated: one involving the entire EA domain (AOD-EA), and the other focusing exclusively within KO (AOD-KO), which is the domain covered by GOCI-II. Validation results of both MLE and DNN-based fused AODs are shown in Fig. 8, where columns represent the validation results of MLE and DNN AODs, and rows denote the results for AOD-EA and -KO. Overall, the validation metrics exhibited notable improvement after the MLE fusion process (including bias correction and MLE fusion) relative to the results of individual GEMS and AMI AOD products (Fig. 8a). The MLE fusion significantly enhanced AOD quality (R = 0.888; RMSE = −0.188; MBE = −0.076; 60.6% within EE). In EA, DNN-based fusion outperformed the MLE fusion, with a substantial enhancement in AOD quality (R = 0.905; RMSE = 0.161; MBE = −0.060; 65.6% within EE). The improvement at low AOD contributed to the notable increase in the percentage of AOD values within the EE. Furthermore, DNN fusion seemed to improve the underestimation of GEMS and AMI AOD at high aerosol loading better than MLE fusion. Fused AOD incorporating data from all three satellite instruments (for AOD-KO) are depicted in Fig. 8c, d. Although the impact of high-AOD underestimation by GEMS and AMI influenced the MLE AOD results, the validation metrics were notably superior to those of individual satellite AOD products including GOCI-II (R = 0.911; RMSE = 0.113; MBE = −0.047; 73.3% within EE). By merging the original AOD dataset according to retrieval error compared to AERONET in different retrieval conditions (NDVI, observation time, aerosol loading and type), the MLE fusion approach thus effectively accommodated nonlinearity in retrieval uncertainty, despite possibly not capturing all complexity in the data. DNN-KO yielded more improved outcome (R = 0.912; RMSE = 0.102; MBE = −0.028; 78.2% within EE). As was in the validation of DNN-EA, the better result of DNN AOD comes from improvement in high AOD (AOD >0.5). Further incorporation of relevant information may enhance the performance of DNN AOD products, but such considerations are beyond the scope of this study. Validation results of the original AOD products and fused AOD in the separate regions of -EA and -KO are listed in Table 5 and Table 6.

## 4.2.2 Aerosol product evaluation

The prognostic error (or, uncertainty) evaluation methodology was based on the framework of Sayer et al. (2020). A comparison of $1\sigma$ retrieval error ($68^{th}$ percentile of absolute AOD error against AERONET, $|\Delta_S|_{68}$; 1σ of Gaussian distribution) according to AOD is shown in Fig. 9. The $|\Delta_S|_{68}$ values represent estimates of the AOD products' uncertainty. Fig 9a shows a prognostic error evaluation of AOD products in East. Asia (AOD-EA). At low AOD, the three original satellite AOD products (GEMS, AMI–MRM, AMI–ESR) displayed large uncertainty that is decreasing rapidly as AOD increases to 0.1. This implies that at low AOD values, the satellite AOD products had relatively high uncertainties, which can likely be attributed to weak aerosol signals and/or cloud contamination issues. With increasing aerosol loading, the uncertainty gradually increases. Among the three original AOD products in EA, GEMS AOD demonstrated the largest uncertainty, which was 0.708 at AOD of 0.638. Uncertainty of the GEMS 550 nm AOD was even larger at 550 nm due to underestimation of AOD and error due to the extrapolation from 443 nm. The prognostic error of GEMS AOD at 550 nm in -EA region was estimated as 0.03 + 0.82AOD. The AMI–ESR AOD had lower uncertainty compared to GEMS AOD, and the AMI–MRM AOD products showed slightly larger uncertainty than AMI–ESR AOD. The prognostic error estimates of AMI–MRM and –ESR AOD product were 0.07 + 0.25AOD and 0.08 + 0.20AOD, respectively. The two fused AOD products in EA (triangle markers, Fig. 9a) indicate that aerosol fusion effectively reduced AOD uncertainty at lower aerosol loadings. As the loading increased, the uncertainties of both MLE-EA and DNN-EA AOD products showed similar retrieval errors as AOD increases to ~0.3. At higher aerosol loading, the MLE-EA AOD showed uncertainties in between GEMS AOD and AMI AODs. Meanwhile, uncertainties of the DNN-EA were even lower than AMI–ESR. The prognostic error estimates for MLE AOD and DNN AOD were 0.02 + 0.43 AOD and 0.05 + 0.23 AOD, respectively. AOD products in -KO region generally showed lower uncertainty than in -EA region (Fig. 9b). Among the four original AOD products, GOCI-II showed lowest uncertainty at high aerosol loading. In the KO, the two AMI AOD products had similar results. This seems to be because the effect of difference in surface reflectance estimation diminishes as the -KO domain contains AERONET sites (e.g., Beijing) with frequent severe haze events. However, relatively higher uncertainty at low aerosol loading still remains for AOD <0.1, even for GOCI-II. The prognostic error estimates for GEMS, GOCI-II, AMI–MRM, and AMI–ESR AOD products are 0.01 + 0.71AOD, 0.05 + 0.17AOD, 0.05 + 0.21AOD, and 0.06 + 0.19AOD, respectively. The MLE-KO AOD showed similar uncertainties with AMI AOD products at high aerosol loading with uncertainty estimate of 0.02 + 0.28AOD. However, the slope of the prognostic error was higher than AMI–MRM AOD because of lower uncertainty at low aerosol loading. Meanwhile, the DNN AOD-KO had the lowest uncertainties among all AOD products with uncertainty estimate of 0.03 + 0.18AOD.

Means and standard deviations of the normalized error ($\Delta_N$) of AOD and fused AOD products are shown in Fig. 10, where the AOD error was normalized using Eq. (6):

$$\Delta_N = \frac{\Delta_S}{\epsilon_T} = \frac{\tau_S - \tau_A}{\sqrt{\epsilon_S^2 + \epsilon_A^2}} \cong \frac{\tau_S - \tau_A}{|\epsilon_S|} \qquad\qquad (6)$$

where $\epsilon_T$ denotes a total expected discrepancy; i.e., a root of the squared sum of expected discrepancies of satellite and AERONET. Values of $\Delta_N$ with a Gaussian distribution of zero mean ($\mu_{\Delta N}$) and unity standard deviation ($\sigma_{\Delta N}$) implies that satellite AODs were calculated appropriately with perfectly characterized errors (Sayer et al., 2020). Here, the satellite expected discrepancy ($\epsilon_S$) assumed to be the EE of MODIS DT land retrieval. Therefore, points near the intersection of mean zero and variance unity lines (Fig. 10) imply that the accuracy and precision of AOD product uncertainty can be explained by the EE of the MODIS DT.

Means and standard deviations of the normalized error of each AOD product collocated at four different AERONET sites are plotted in Fig. 10. Beijing_RADI and Anmyon sites represent polluted atmosphere over land and coast, respectively. Beijing is one of the largest cities in East Asia, and Anmyon is located near the Yellow Sea, over which long-range aerosol transport passes (Lee et al., 2019). AERONET sites remote from large cities in Japan, TGF_Tsukuba and Okinawa_Hedo were selected to demonstrate results for relatively clear atmosphere over land and coast, respectively. Over the polluted land site, the original AOD products showed negative bias. AMI–MRM, AMI–ESR, and GOCI-II AOD had $\sigma_{\Delta N}$ close to unity, while that of the GEMS AOD was higher. Both fused AOD products had smaller $\sigma_{\Delta N}$ values than unity, meaning that the fused AOD products have higher precision than MODIS DT over polluted atmosphere. Among the two fused AOD, DNN showed better performance with better accuracy ($\mu_{\Delta N}$ closer to zero) and higher precision (lower $\sigma_{\Delta N}$), due to better estimation of high AOD as shown in discussions regarding Fig. 8. Over clear land site, among original AOD products, GEMS AOD had highest precision and GOCI-II AOD had highest accuracy. Both AMI AOD products had low precision, with positive bias of AMI–MRM AOD and negative bias of AMI–ESR AOD. The MLE AOD showed improvement in precision but had positive bias after fusion. This seems to be because the bias correction procedure is applied regardless of AOD value. For the case of GEMS AOD, small bias over clear atmosphere may be overcorrected to have positive bias before MLE fusion. DNN AOD does not seem to have such a problem, with even better precision than at the polluted site. AMI AOD collocated at the Anmyon site had a strong positive bias (Fig. 10b) due to overestimation of AOD over turbid water. Over clear coastal areas, AMI–MRM, AMI–ESR and GOCI-II AOD products displayed relatively high precision with positive bias. Cloud contamination may cause the positive bias. Over both coastal sites, MLE-KO AOD outperformed DNN-KO AOD. Relatively consistent retrieval conditions of ocean surface than land surface may lead to better quantification of uncertainty for MLE fusion.

### 4.2.3 Diurnal variation during aerosol transport in East Asia

A case of long-range aerosol transport over the Yellow Sea is described in Fig. 11. A scattering aerosol plume originating from the Shandong Peninsula was transported toward Japan, penetrating South Korea. Three AERONET sites were chosen to assess diurnal variations of AOD during this event. The diurnal variations of AERONET AOD were similar at Yonsei_University (37.6 °N, 126.9 °N) and KORUS_UNIST_Ulsan (35.6 °N, 129.2 °E) (Fig. 11b–c), characterized by a peak at around 01:30 UTC followed by a gradual decline throughout the day. At the Yonsei_University site (Fig. 11b), AERONET AOD data for the 00–02 UTC period was absent due to cloud cover. GEMS AOD exhibited a diurnal pattern

similar to that of AERONET but, as established in earlier analyses, GEMS tends to underestimate AOD, particularly with high aerosol loadings. AMI–MRM AOD captured AERONET AOD well until 02 UTC, after which it underestimated AOD. AMI–ESR AOD displayed similar trends but with the most negatively biased retrieval among all AOD products, consistent with the negative bias observed in error analysis (Fig. 7d, h). GOCI-II AOD demonstrated the most accurate performance during this event. After MLE fusion, MLE AOD closely followed the diurnal AOD variation of AERONET, with slight underestimation. Due to the greater uncertainty of GEM AOD under high aerosol loadings, its weight in the MLE fusion was lower than those of other satellite AOD products, with MLE AOD being more similar to other AOD products. The MLE fusion method successfully combined each AOD product by considering retrieval uncertainties under various conditions. DNN AOD exhibited a closer value of AOD compared to MLE AOD, as expected based on former discussions. However, with DNN AOD, diurnal AOD variation may be misunderstood to be consistently increasing after 04 UTC, while the AERONET AOD is decreasing again after 05 UTC (Fig. 11b). Collocated results from the Gosan_SNU site (33.3 °N, 126.2 °E) (Fig. 11d), which was located outside the main plume, had a moderately high aerosol loading with an average AERONET AOD of ~0.4. GEMS consistently underestimated AOD throughout the day, whereas both AMI–AOD products overestimated AOD slightly in the early morning (00–02 UTC). Both MLE and DNN AOD showed almost the same AOD values throughout the day at Gosan_SNU site.

## 5 Conclusion

Individual AOD products from GEMS, AMI, and GOCI-II were validated over the period of November 2022 to April 2023 using AERONET level 1.5 AOD as a reference. Linear regression line of GEMS AOD and AERONET AOD exhibited a slope of 0.368, indicating underestimation of high aerosol loading relative to AERONET AOD. The GOCI-II YAER algorithm yielded better performance than GEMS and AMI. Within AMI YAER algorithms, the validation of AMI–ESR AOD was slightly more accurate than that of AMI–MRM AOD due to the better estimation of urban surface reflectance in the former. Two AOD data fusion methods were developed using the same input variables—GEMS, AMI (–MRM and –ESR), and GOCI-II AOD—with NDVI, observation time, and selected aerosol type from each algorithm. For MLE fusion, pixel-level biases and RMSEs of aerosol algorithms were calculated by comparing individual satellite AODs with level 2.0 AERONET AODs for November 2021 to October 2022, with this period being used also for DNN model training. Fusion outcomes were categorized into two groups based on the individual AODs used in fusion and then evaluated. The fused AOD-EA from both MLE and DNN-based fusion yielded better results relative to GEMS and AMI AOD products. DNN AOD outperformed MLE AOD, particularly in terms of quantifying AOD at high aerosol loading. Due to small spatial domain of the GOCI-II observation, fused AOD-KO was selected for evaluation of fusion involving GOCI-II AOD. Both MLE and DNN AOD-KO yielded better results than GOCI-II AOD. MLE AOD retained underestimation owing to the GEMS tendency to underestimate high aerosol loadings. This issue was not observed with DNN AOD-KO. Evaluation of AOD bias with respect to observation time indicated that both fusion algorithms stabilized diurnal error variations,

suggesting that fusion AOD enhances our understanding of the hourly evolution of aerosol distributions. The performance of each AOD product was assessed by comparing prognostic errors. At lower aerosol loadings, the fused AOD products yielded low uncertainties, overcoming large uncertainty of individual AOD products. The MLE AOD uncertainty increased sharply with aerosol loading; DNN AOD did not display such a behavior. Prognostic error analysis revealed that DNN-KO yielded the best performance, with lower uncertainty. A case of long-range aerosol transport was chosen for diurnal monitoring. MLE fusion, accounting for retrieval uncertainty from each aerosol algorithm, improved the hourly AOD distribution when compared with AERONET AOD. DNN AOD tracked AERONET AOD closely, yielding AOD estimates that were more closely aligned with AERONET values. The performance of aerosol data fusion can be improved with more dataset in the future study. For the MLE fusion, more sample leads to better representativeness of uncertainty weight. On the other hand, more dataset leads to better train performance of the DNN model. Moreover, DNN model in the future study will include more variables to predict optimal AOD. In April 2023, the US National Aeronautics and Space Administration launched the next series of global geostationary environmental constellation instruments, TEMPO; the European Space Agency launched FCI in December 2022 and is planning to launch Sentinel-4 in 2025; and the NOAA GEOstationary eXtended Observations (GeoXO) satellite system is planned to form a constellation of geostationary satellite instruments. The application of aerosol fusion described here to these geostationary satellite projects may enable global production of high-quality aerosol data.

*Code availability.* The statistical and DNN-based aerosol fusion codes are available on request.

*Data availability.* The statistical and DNN-based aerosol fusion data are available on request.

*Author Contributions.* MK, HL, SG and JK designed the experiment. MK carried out the data processing. SL and YC provided support with the data. MK wrote the manuscript, with contributions from all co-authors. JK reviewed and edited the article. JK provided support and supervision. All authors analyzed the measurement data and took part in manuscript preparation.

*Competing Interests.* The contact author has declared that none of the authors have any competing interests.

*Acknowledgements.* We thank all principal investigators and their staff for establishing and maintaining the AERONET sites used in this investigation. The authors acknowledge the National Meteorological Satellite Center, the Korea Institute of Ocean Science and Technology, and the National Institute of Environmental Research for the satellite data. The work of Jhoon Kim was supported by the Yonsei Fellow Program, funded by Lee Youn Jae and Samsung Advanced Institute of Technology (SAIT). This work was supported by the National Research Foundation of Korea (NRF) grant funded by the Korea government (MSIT) (RS-2024-00346149).

*Financial Support.* This work was supported by a grant from the National Institute of Environment Research (NIER), funded by the Ministry of Environment (MOE) of the Republic of Korea (NIER-2023-04-02-082).

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

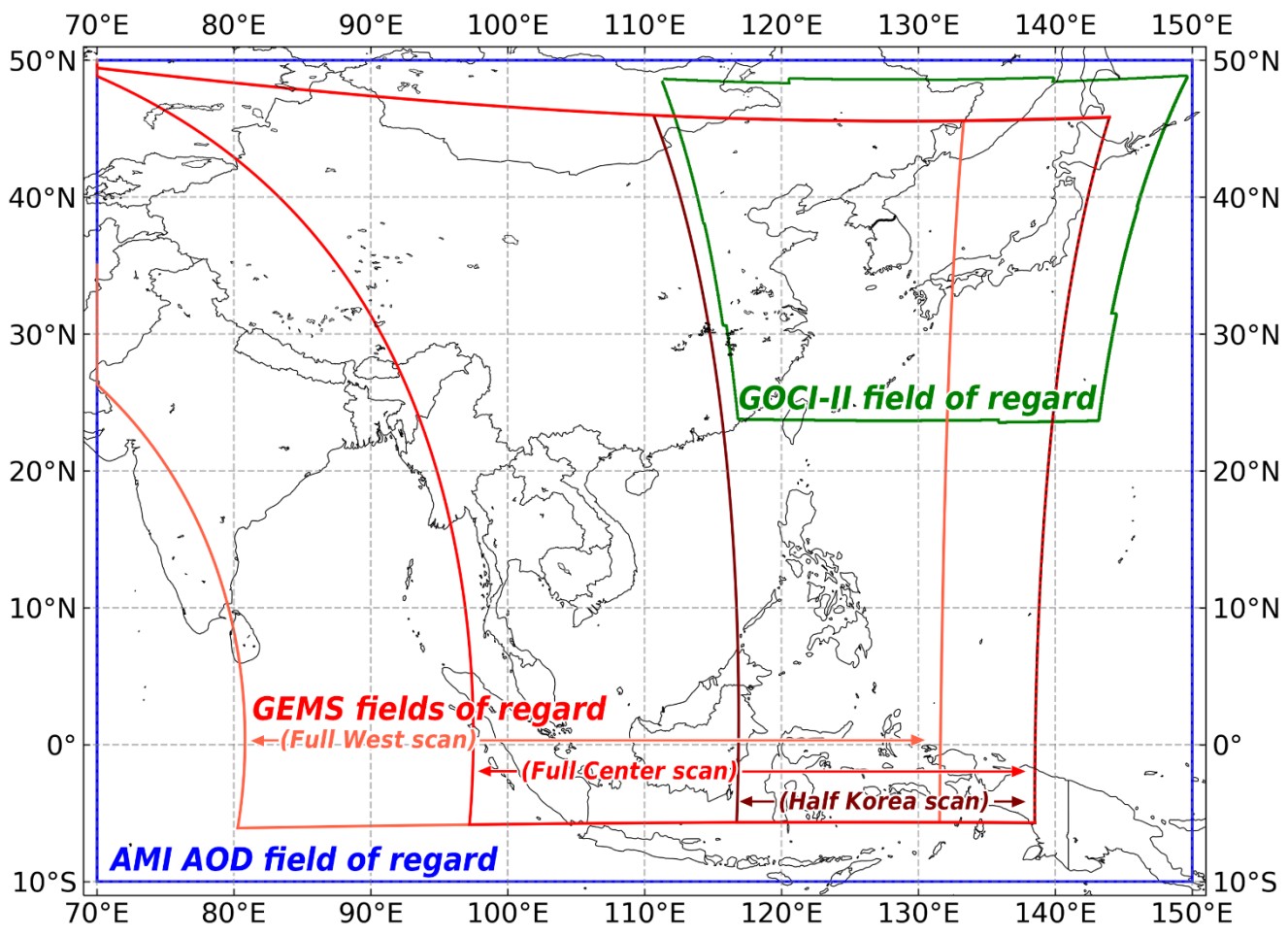

**Figure 1. Fields of regard for GEMS, AMI, and GOCI-II. AMI AOD were retrieved only within the 70°E–150°E, 10°S–50°N area to match the GEMS field of regard.**

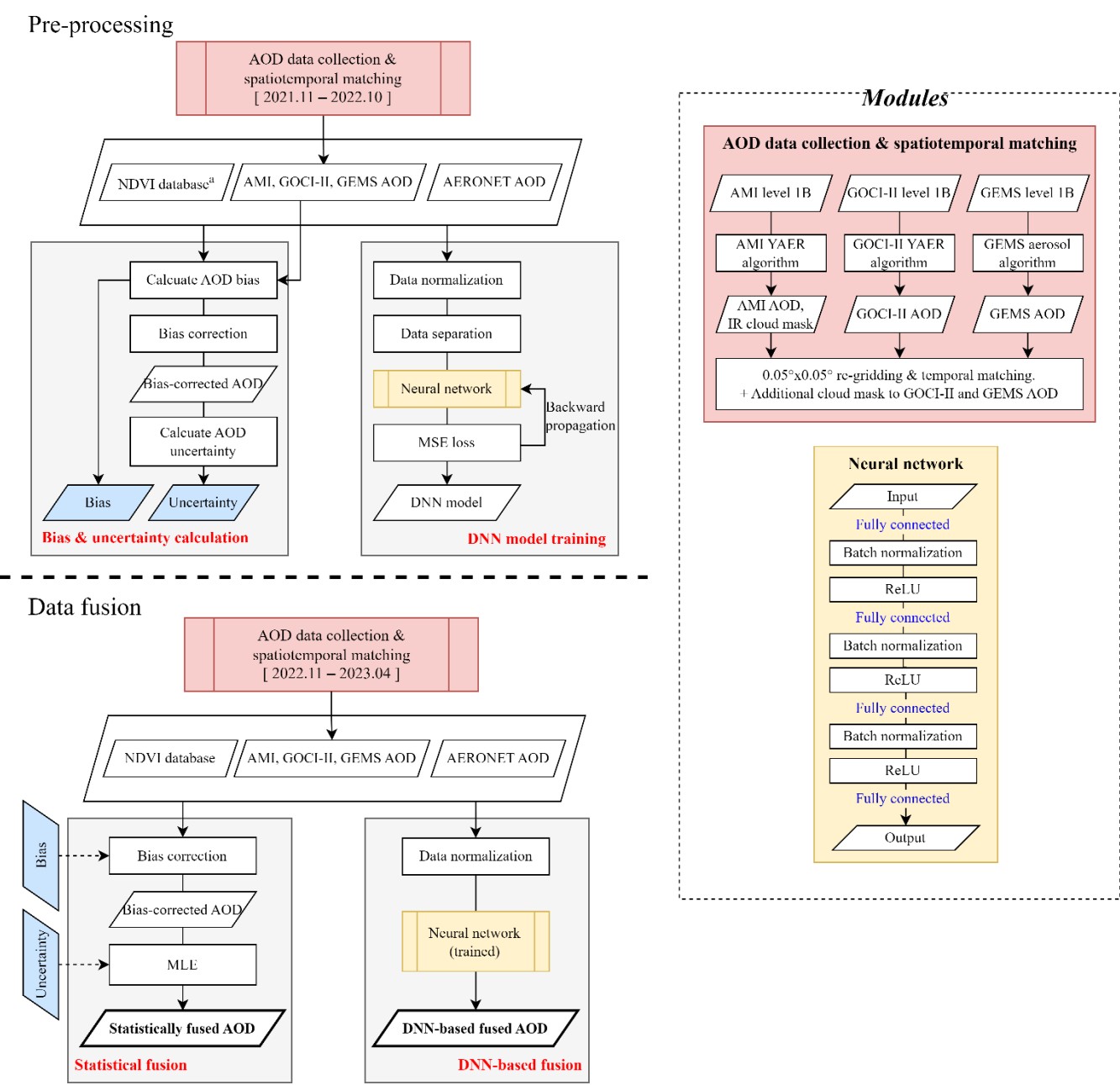

**Figure 2. Flowchart of the fusion algorithm. Red- and yellow-colored boxes represent commonly used modules for both preprocessing and data fusion procedures. The Normalized Differential Vegetation Index (NDVI) database[a] was generated using AMI Minimum Reflectance Method (MRM) surface reflectance in green, red, and Near-Infrared (NIR) channels.**

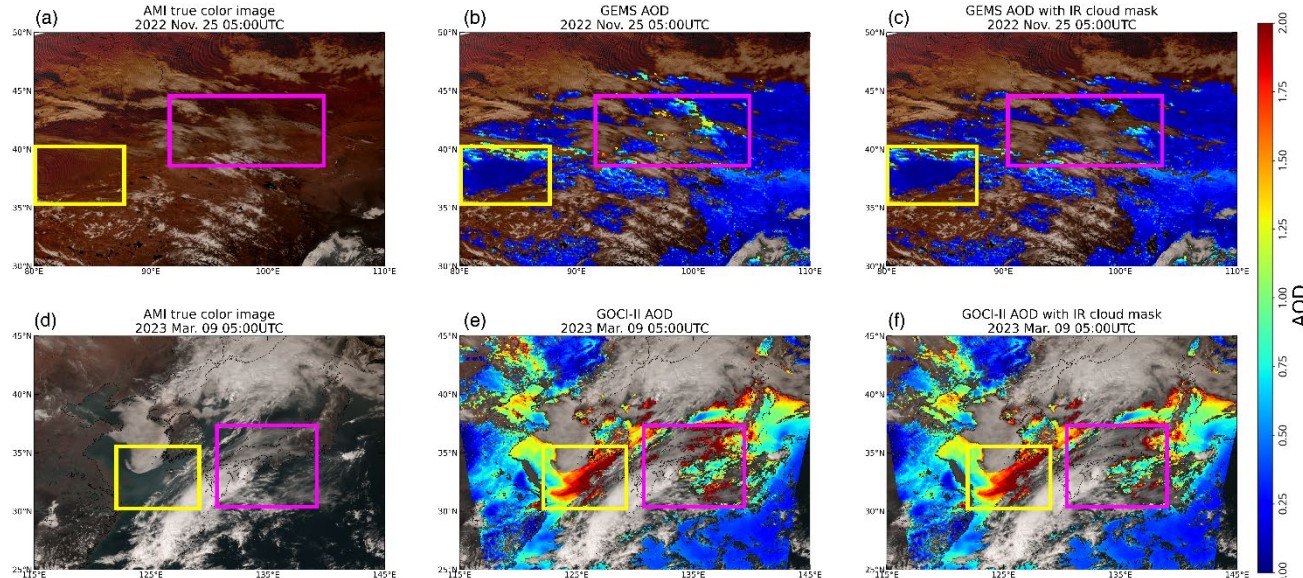

**Figure 3. Examples of additional Infrared (IR) cloud masking for GEMS and GOCI-II AOD. (a–c) A case of GEMS AOD on 25 November 2022; (d–f) A case of GOCI-II AOD on 09 March 2023. Each column shows an AMI true color image, original AOD, and AOD with additional cloud mask of corresponding cases. Yellow boxes correspond to cloud-free zones; magenta boxes correspond to areas in which GEMS or GOCI-II misidentified cloud pixels.**

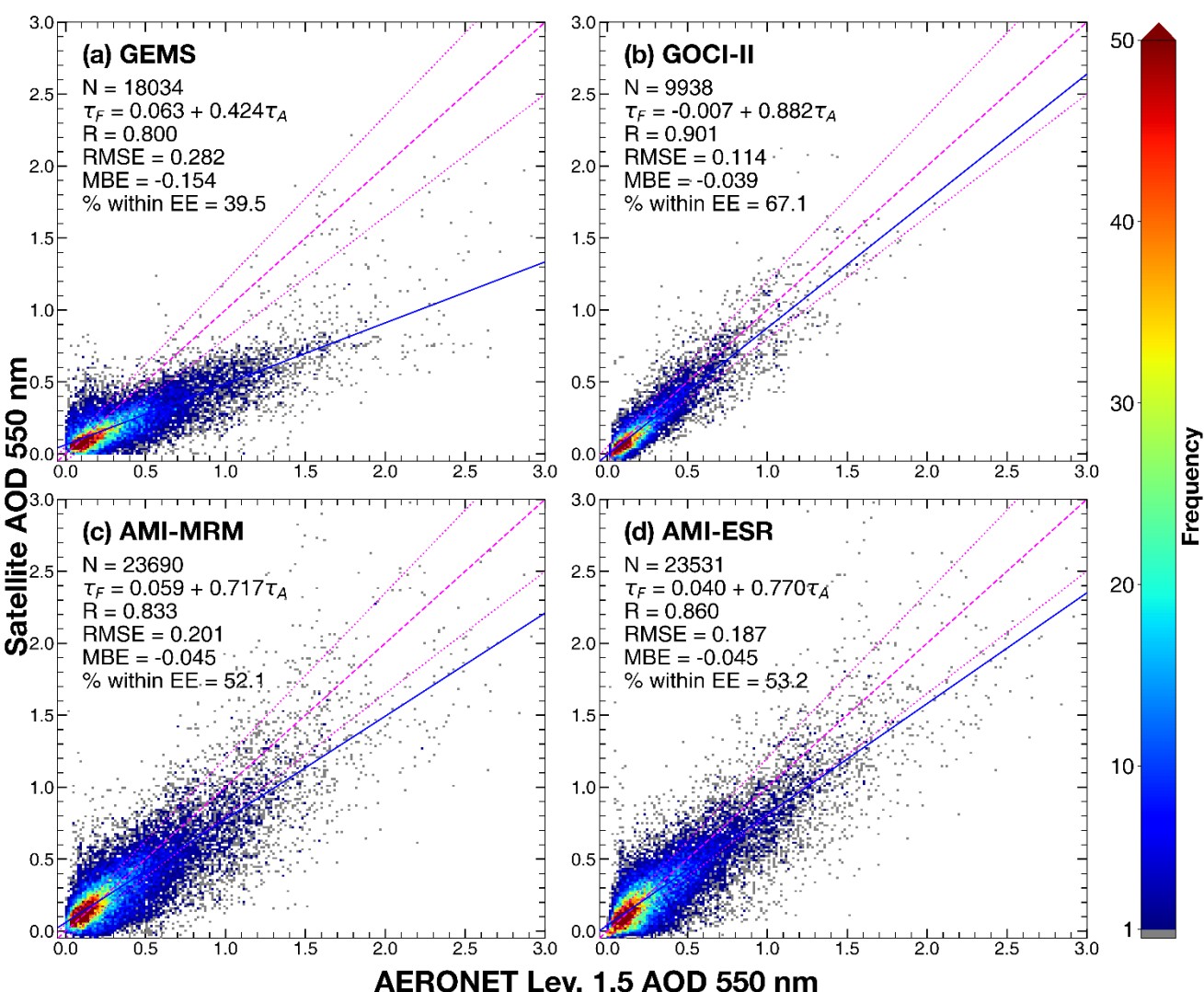

**Figure 4. 2-dimensional histograms of AERONET AOD vs. GEMS (a), GOCI-II (b), AMI–MRM (c), and AMI–ESR AOD (d)**
**frequencies. The number of collocated points (N), linear regression equations, Pearson's correlation coefficient (R), root mean squared errors (RMSE), mean bias errors (MBE), and percentage within the expected error envelope (% within EE; EE: $\pm(0.05+0.15\ \tau_A)$) is shown. Dashed line and dotted lines indicate one-to-one line and expected error envelopes. Blue line indicates linear regression line of the satellite AOD and AERONET AOD**

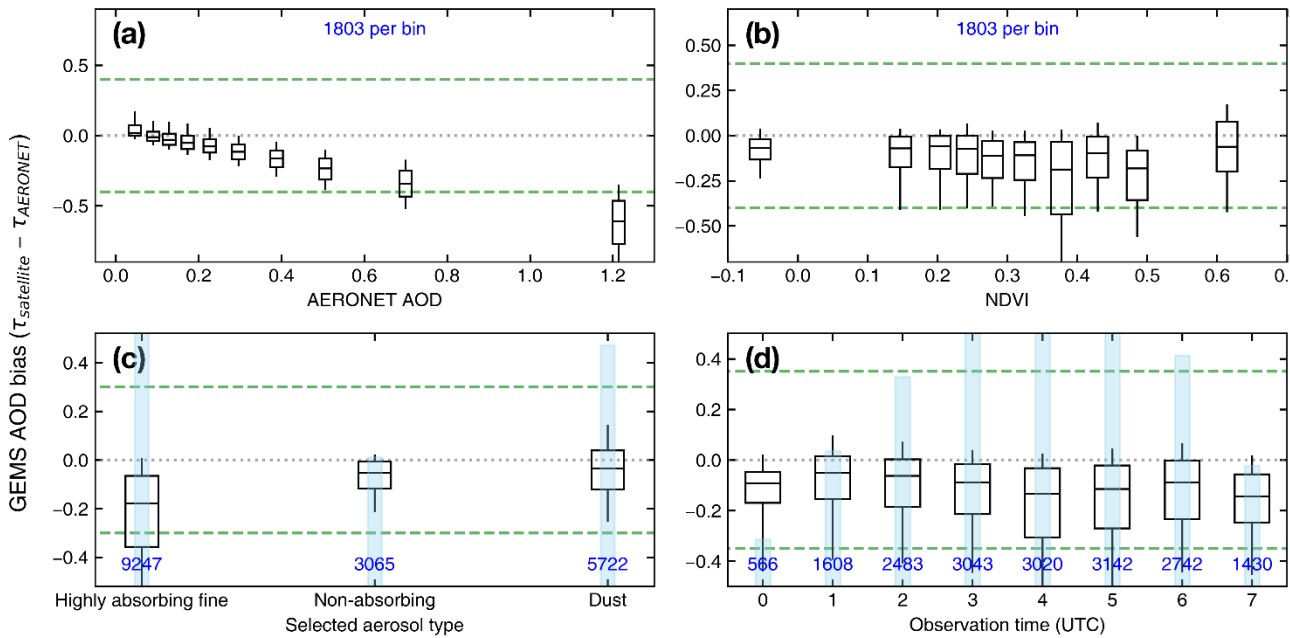

**Figure 5. AOD bias of GEMS AOD relative to AERONET AOD (a), NDVI (b), selected aerosol type (c), and observation time (d). Whisker ends correspond to the 10th and 90th percentiles of the bin. Box ends correspond to the 25th and 75th percentiles. Horizontal lines in each box indicate bin median. Green dashed line indicates the y-axis range of GOCI-II AOD in corresponding panels. Numbers and bar plots in blue indicate the number of collocated AOD points in each box–whisker.**

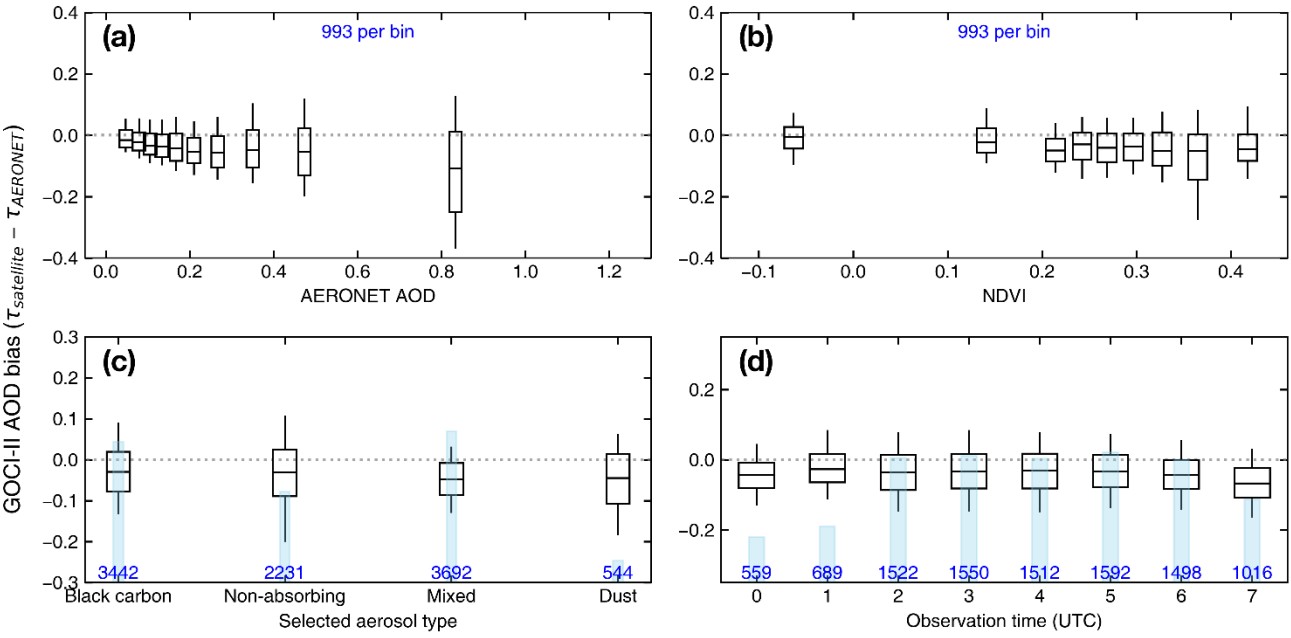

**Figure 6. As for Fig. 5, but for GOCI-II AOD.**

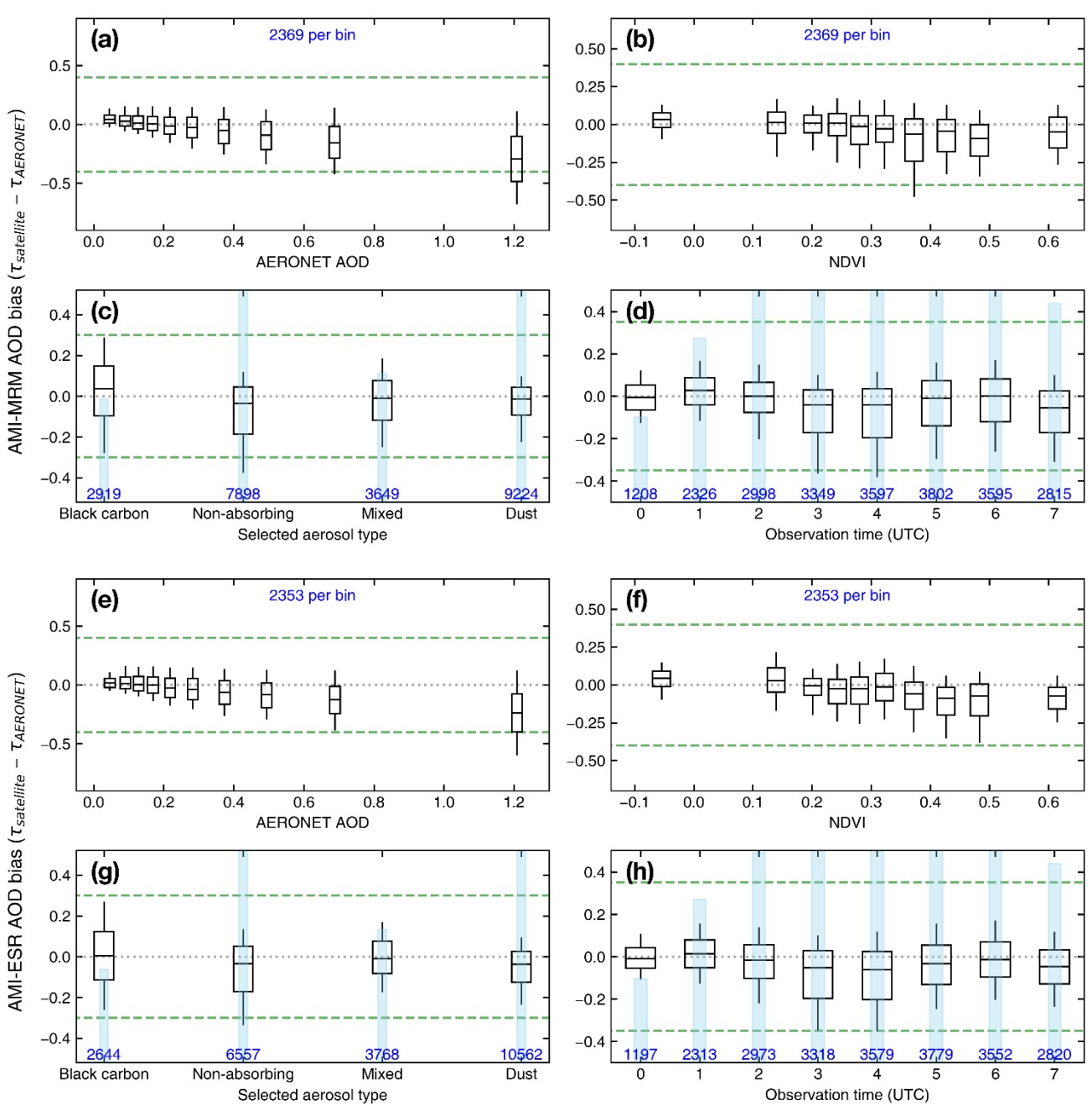

 **Figure 7. As for Fig. 5, but for AMI–MRM and AMI–ESR AOD.**

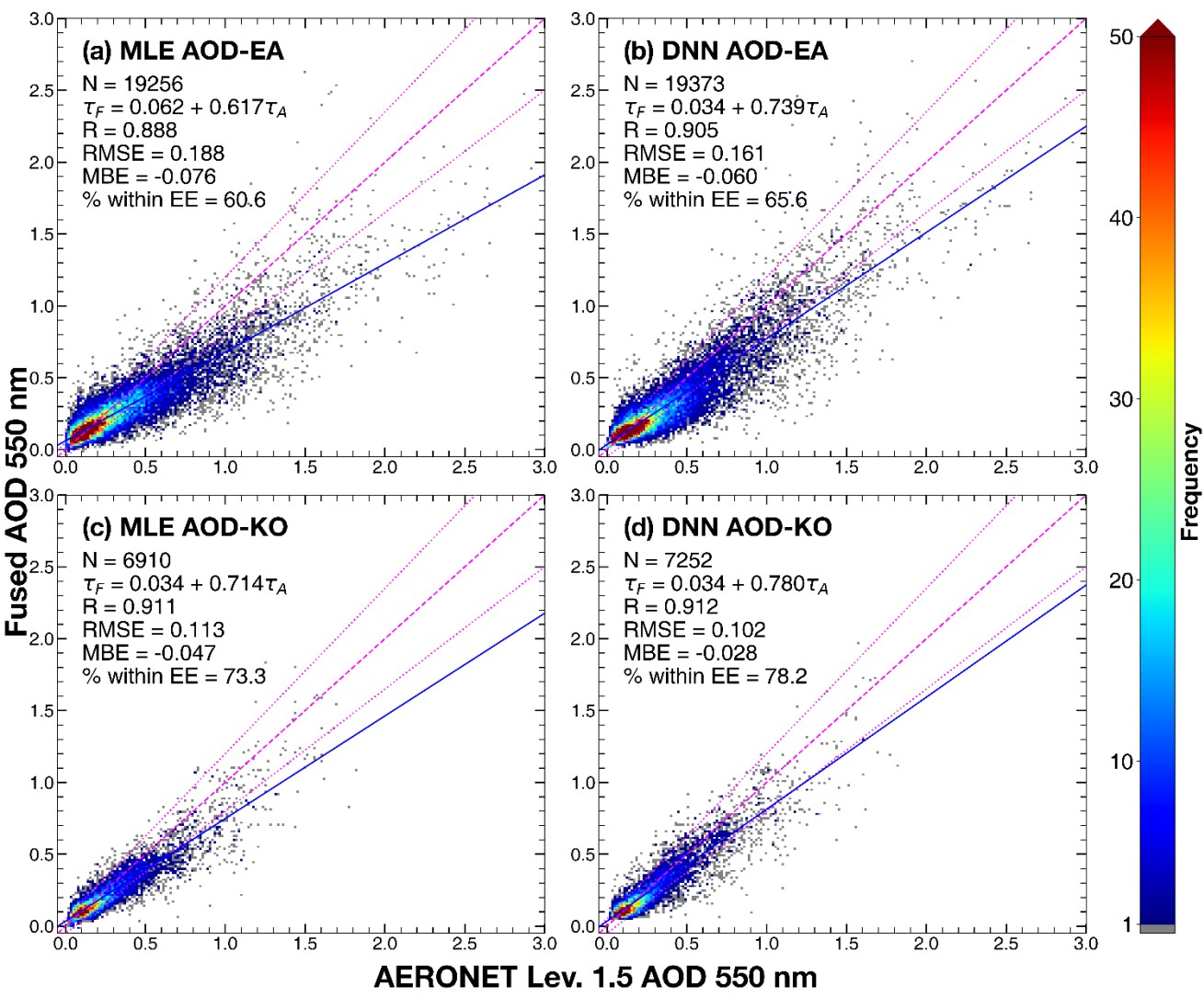

**Figure 8. As for Fig. 4, but for Maximum Likelihood Estimation (MLE) AOD-EA (a), Deep Neural Network (DNN) AOD-EA (b), MLE AOD-KO (c), and DNN AOD-KO (d).**

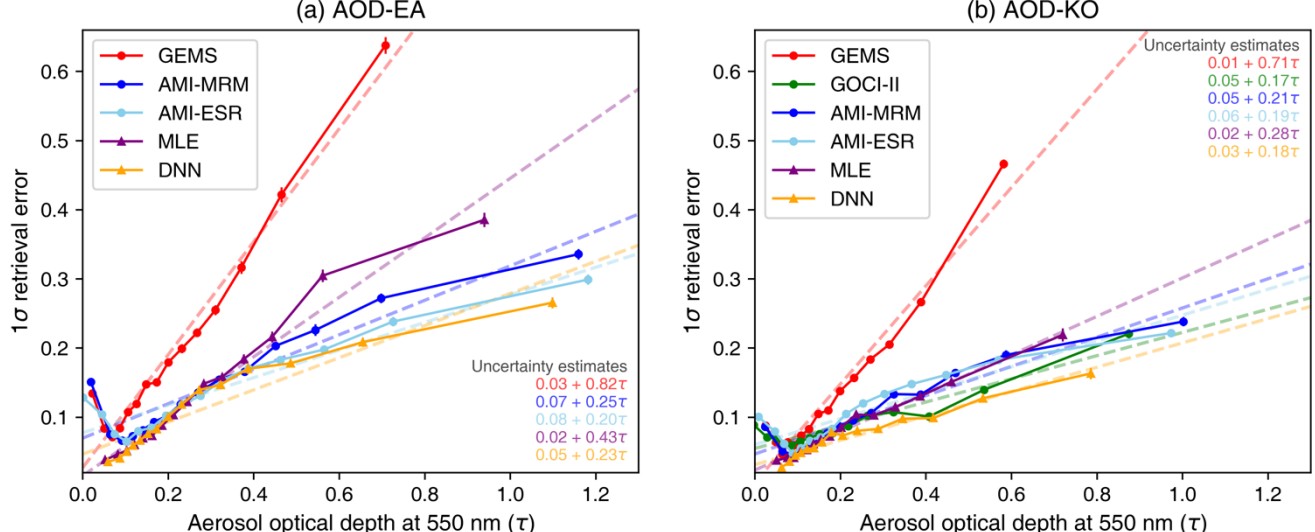

**Figure 9. Changes in uncertainty of AOD products with increasing AOD. 68th percentile value of retrieval error is defined as 1σ retrieval error and plotted against AOD value. Vertical lines of each marker represent the difference of 67th and 69th percentiles, thus indicating the error of 1σ retrieval error. Dashed lines show linear regression (uncertainty estimates) of 1σ retrieval error.**

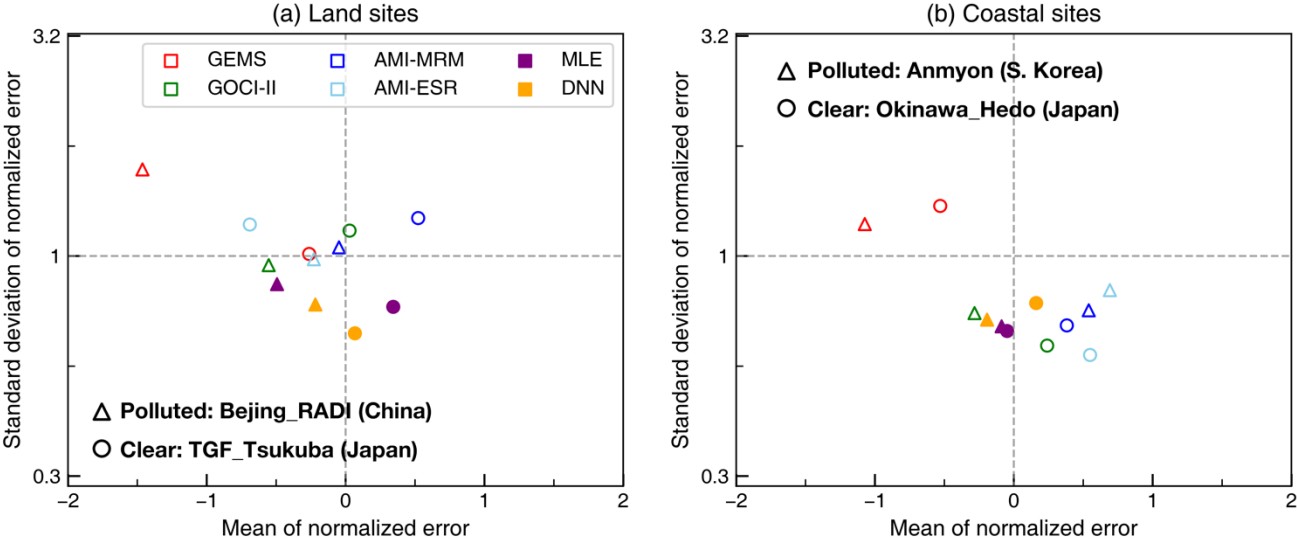

**Figure 10. Mean and standard deviation of normalized error for each AOD product collocated at selected land (a) and coastal (b) AERONET sites. The Bejijing_RADI and TGF_Tsukuba sites were chosen to represent polluted atmosphere over land and ocean, respectively. The Anmyon and Okinawa_Hedo sites were chosen to represent clear atmosphere over land and ocean, respectively.**

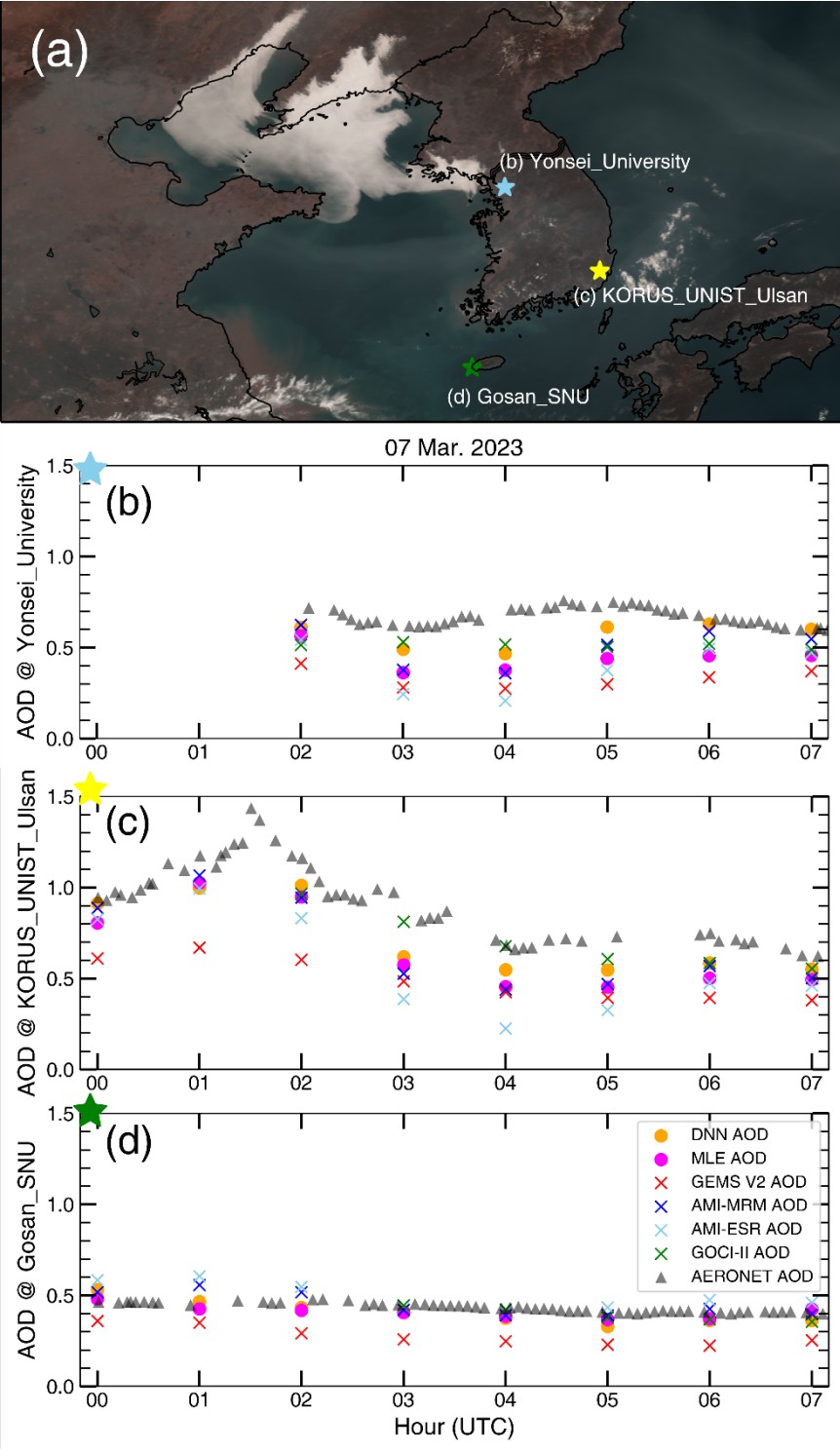

**Figure 11. A case of aerosol transport over the Korean Peninsula on 07 March 2023. (a) AMI true color image at 04 UTC. (b)–(d) correspond to AERONET sites marked in (a). Gray triangles indicate AERONET AOD; original satellite AOD products are**

820 indicated by × symbols in different colors; fused AOD products around the Korean Peninsula (AOD-KO) are indicated by circles in different colors.

Table 1: Specification of instruments in Geo Kompsate-2 (GK-2) mission.

| Satellite | GK-2A | GK-2B | |
|---|---|---|---|
| Payload | AMI | GOCI-II | GEMS |
| Channels | 16 | 14 | 1,024 |
| Spatial resolution of radiance | 0.5 km (red), 1 km (VIS), 2 km (IR) | 0.25 km | 3.5 km × 7.7 km |
| Temporal resolution | 10 min (full-disk scan) | 1 h | 1 h |
| Wavelength range | 0.4–13 μm | 375–860 nm | 300–500 nm |
| FWHM | 10–20 nm | 10–20nm | 0.6 nm |
| Launch | December 2018 | February 2020 | |
| Lifetime | 10 years | | |
| Location | 128.2° | | |
| Details of aerosol products | | | |
| Aerosol algorithm | AMI YAER algorithm (Kim et al., 2024) | GOCI-II YAER algorithm (Lee et al., 2023) | GEMS AERAOD retrieval algorithm (Cho et al., 2023) |
| Spatial resolution of aerosol product | 6 km | 2.5 km | 3.5 km × 7.7 km |
| Land surface reflectance estimation | MRM & Estimation from SWIR channel | Minimum reflectance method (MRM) | |
| Ocean surface reflectance estimation | Cox & Munk method (Cox and Munk, 1954) | | MRM |
| Inversion channels | VIS-NIR 4 bands | UV-NIR 12 bands | UV-VIS 6 spectrally binned bands |
| Algorithm version | Research algorithm | Ver. 1.1 | Ver. 2.0 |

**Table 2 The AERONET site information used in this study. All sites had valid collocation at level 1.5 with spaceborne AOD product from November, 2022 to April, 2023. Site names with asterisk (*) refers to sites where level 2.0 data from November, 2021 to October, 2022 are used for pre-processing.**

| # | Site name | Longitude (°E) | Latitude (°N) | Elevation (m) | | # | Site name | Longitude | Latitude | Elevation |
|---|---|---|---|---|---|---|---|---|---|---|
| 1 | AOE_Baotou | 109.629 | 40.852 | 1314 | | 46 | *Gwangju_GIST | 126.843 | 35.228 | 52 |
| 2 | *ARIAKE_TOWER | 130.272 | 33.104 | 15 | | 47 | *Hankuk_UFS | 127.266 | 37.339 | 167 |
| 3 | *Amity_Univ_Gurgaon | 76.916 | 28.317 | 285 | | 48 | *Hokkaido_University | 141.341 | 43.075 | 59 |
| 4 | *Anmyon | 126.33 | 36.539 | 47 | | 49 | *Hong_Kong_PolyU | 114.18 | 22.303 | 30 |
| 5 | *BMKG_GAW_PALU | 120.183 | -1.65 | 1370 | | 50 | *Hong_Kong_Sheung | 114.117 | 22.483 | 40 |
| 6 | Baengnyeong | 124.63 | 37.966 | 136 | | 51 | *IIT_Delhi | 77.193 | 28.545 | 15 |
| 7 | *Bangkok | 100.518 | 13.749 | 57 | | 52 | *Jaipur | 75.806 | 26.906 | 450 |
| 8 | Beijing-CAMS | 116.317 | 39.933 | 106 | | 53 | *Jambi | 103.642 | -1.632 | 30 |
| 9 | Beijing | 116.381 | 39.977 | 92 | | 54 | *KORUS_UNIST_Ulsan | 129.19 | 35.582 | 106 |
| 10 | Beijing_PKU | 116.31 | 39.992 | 53 | | 55 | *Kanpur | 80.232 | 26.513 | 123 |
| 11 | Beijing_RADI | 116.379 | 40.005 | 59 | | 56 | *Kaohsiung | 120.292 | 22.676 | 15 |
| 12 | *Bhola | 90.756 | 22.227 | 7 | | 57 | *Kemigawa_Offshore | 140.023 | 35.611 | 8 |
| 13 | Bidur | 85.14 | 27.895 | 576 | | 58 | *Lahore | 74.264 | 31.48 | 209 |
| 14 | *Bukit_Kototabang | 100.318 | -0.202 | 864 | | 59 | *Lulin | 120.874 | 23.469 | 2868 |
| 15 | *Cape_Fuguei_Station | 121.538 | 25.297 | 40 | | 60 | *Makassar | 119.572 | -4.998 | 16 |
| 16 | *Chachoengsao | 101.45 | 13.5 | 60 | | 61 | Mandalay_MTU | 96.186 | 21.973 | 104 |
| 17 | Chen-Kung_Univ | 120.205 | 22.993 | 50 | | 62 | *Manila_Observatory | 121.078 | 14.635 | 63 |
| 18 | Chiang_Dao | 98.961 | 19.455 | 450 | | 63 | *NAM_CO | 90.962 | 30.773 | 4746 |
| 19 | *Chiang_Mai_Met_Sta | 98.972 | 18.771 | 312 | | 64 | *ND_Marbel_Univ | 124.843 | 6.496 | 70 |
| 20 | *Chiba_University | 140.104 | 35.625 | 60 | | 65 | *Niigata | 138.942 | 37.846 | 10 |
| 21 | *DRAGON_Hakuba | 137.864 | 36.701 | 703 | | 66 | *Nong_Khai | 102.717 | 17.877 | 175 |
| 22 | *DRAGON_Iida | 137.842 | 35.517 | 490 | | 67 | *Noto | 137.137 | 37.334 | 200 |
| 23 | *DRAGON_Ina | 137.961 | 35.847 | 683 | | 68 | Okinawa_Hedo | 128.249 | 26.867 | 60 |
| 24 | *DRAGON_Kofu | 138.572 | 35.679 | 314 | | 69 | *Osaka | 135.591 | 34.651 | 50 |
| 25 | *DRAGON_Matsumoto | 137.978 | 36.251 | 626 | | 70 | *Palangkaraya | 113.946 | -2.228 | 27 |
| | | | | | | 71 | *Pokhara | 83.975 | 28.187 | 800 |

| 26 | *DRAGON_Minowa | 137.981 | 35.915 | 713 | 72 | *Pontianak | 109.191 | 0.075 | 2 |
|----|---------------|---------|--------|-----|----|-----------|---------|-------|-----|
| 27 | *DRAGON_Mt_Happo | 137.798 | 36.697 | 1846 | 73 | *QOMS_CAS | 86.948 | 28.365 | 4276 |
| 28 | *DRAGON_Mt_Haruna | 138.878 | 36.475 | 1359 | 74 | *Seoul_SNU | 126.951 | 37.458 | 116 |
| 29 | *DRAGON_Mt_Krigamine | 138.168 | 36.098 | 1674 | 75 | Shirahama | 135.357 | 33.693 | 10 |
| 30 | *DRAGON_Omachi | 137.851 | 36.503 | 751 | 76 | Silpakorn_Univ | 100.041 | 13.819 | 72 |
| 31 | *DRAGON_Suwa | 138.109 | 36.046 | 766 | 77 | *Singapore | 103.78 | 1.298 | 30 |
| 32 | *DRAGON_Takayama | 137.304 | 36.253 | 1296 | 78 | *Socheongcho | 124.738 | 37.423 | 28 |
| 33 | *Dalanzadgad | 104.419 | 43.577 | 1470 | 79 | Songkhla_Met_Sta | 100.605 | 7.184 | 15 |
| 34 | *Dhaka_University | 90.398 | 23.728 | 34 | 80 | *Sorong | 131.268 | -0.875 | 127 |
| 35 | *Dibrugarh_Univ | 94.897 | 27.451 | 119 | 81 | *Sra_Kaeo | 102.504 | 13.689 | 68 |
| 36 | *Doi_Ang_Khang | 99.045 | 19.932 | 1536 | 82 | *TASA_Taiwan | 121.001 | 24.784 | 99 |
| 37 | *Dongsha_Island | 116.729 | 20.699 | 5 | 83 | *TGF_Tsukuba | 140.096 | 36.114 | 25 |
| 38 | Douliu | 120.545 | 23.712 | 60 | 84 | *Tai_Ping | 114.362 | 10.376 | 4 |
| 39 | *EPA-NCU | 121.185 | 24.968 | 144 | 85 | *Taipei_CWB | 121.538 | 25.015 | 26 |
| 40 | Erlin | 120.41 | 23.925 | 16 | 86 | *USM_Penang | 100.302 | 5.358 | 51 |
| 41 | *Fukue | 128.682 | 32.752 | 80 | 87 | Ubon_Ratchathani | 104.871 | 15.246 | 120 |
| 42 | *Fukuoka | 130.475 | 33.524 | 30 | 88 | *Ussuriysk | 132.163 | 43.7 | 280 |
| 43 | *Gandhi_College | 84.128 | 25.871 | 60 | 89 | *XiangHe | 116.962 | 39.754 | 36 |
| 44 | *Gangneung_WNU | 128.867 | 37.771 | 60 | 90 | *Xitun | 120.617 | 24.162 | 91 |
| 45 | *Gosan_NIMS_SNU | 126.206 | 33.3 | 52 | 91 | *Yonsei_University | 126.935 | 37.564 | 97 |

**Table 3: Cloud masking tests of AMI Yonsei AErosol Retrieval (YAER) algorithm.**

| Cloud detection test | Threshold | Surface type |
|---|---|---|
| BTD 9 max | <−28 | Only over land |
| BTD 14 max | <−28 | Only over land |
| Reflectance 1.3 | >0.025 | Over both land and ocean |
| BTD 15, 16 | <10 | Over both land and ocean |
| BTD 13, 16 | ≤11 | Over both land and ocean |

830

**Table 4: Different cases of AOD availability in the fusion process and corresponding data fusion strategies.**

| | GEMS | AMI (MRM, ESR) | GOCI-II | MLE fusion | DNN fusion |
|---|---|---|---|---|---|
| AOD availability | ○ | ○ | ○ | Bias-corrected + MLE | Separate DNN model for each case |
| | ○ | ○ | | Bias-corrected + MLE | Separate DNN model for each case |
| | ○ | | ○ | Bias-corrected + MLE | Separate DNN model for each case |
| | | ○ | ○ | Bias-corrected + MLE | Separate DNN model for each case |
| | ○ | | | Bias-corrected | Separate DNN model for each case |
| | | ○ | | Bias-corrected | Separate DNN model for each case |
| | | | ○ | Bias-corrected | Separate DNN model for each case |

**Table 5: Validation statistics of original AOD products and fused AOD products in -EA region. Validation period is from November 2022 to April 2023.**

| | Original AOD products | | | Fused AOD products | |
|---|---|---|---|---|---|
| | GEMS | AMI-MRM | AMI-ESR | MLE AOD | DNN AOD |
| R | 0.800 | 0.834 | 0.860 | 0.888 | 0.905 |
| RMSE | 0.287 | 0.201 | 0.187 | -0.188 | 0.161 |
| MBE | -0.154 | -0.045 | -0.045 | -0.076 | -0.060 |
| % within EE | 39.5 | 52.1 | 53.3 | 60.6 | 65.6 |

835

**Table 6: Validation statistics of original AOD products and fused AOD products in -KO region.**

| | Original AOD products | Fused AOD products |
|---|---|---|

|  | GEMS | AMI-MRM | AMI-ESR | GOCI-II | MLE AOD | DNN AOD |
|---|---|---|---|---|---|---|
| R | 0.807 | 0.878 | 0.867 | 0.901 | 0.911 | 0.912 |
| RMSE | 0.187 | 0.129 | 0.129 | 0.114 | 0.113 | 0.102 |
| MBE | -0.086 | 0.017 | -0.002 | -0.038 | -0.047 | -0.028 |
| % within EE | 51.7 | 63.1 | 58.8 | 67.1 | 73.3 | 78.2 |