# Peer review of "AOD data fusion with Geostationary Korea Multi-Purpose Satellite (GEO-KOMPSAT-2) instruments GEMS, AMI, and GOCI-II: Statistical and deep neural network methods"

_Atmospheric Measurement Techniques, 2023_

## Referee Comment (RC1)

The manuscript entitled by "AOD data fusion with Geostationary Korea Multi-Purpose Satellite (Geo-KOMPSAT) instruments GEMS, AMI, and GOCI-2: Statistical and deep neural network methods" are shown to the data fusion for various AOD retrieval products to enhance accuracy and stability of datasets. This manuscript showed details of statistical fusion methods and pre-processing before data fusion.

For the readability of manuscript, it needs to be checked the English correction. In addition, chapters of the manuscript must be re-constructed. In particular, some words and phrases are unclear. Although this manuscript is highly useful for the application of air quality and climate, the manuscript will be improved before publication. I summarized the details of comments.

Major comments

1) Title: This work used the AOD retrieval from GEMS, AMI, and GOCI-2, which are GK2 mission, not GK mission. I suggested that the title will change 'Geo-KOMPSAT-2 (GK2)'. In addition, the author should check and correct this word in all manuscript.

2) Introduction: This manuscript is purposed on showing the result of AOD data fusion. Although the manuscript explained several previous AOD retrieval algorithms, other product algorithms (such as aerosol index, SSA, ALH) are not essential to explain. Please simplify the introduction session more focusing on the AOD retrieval algorithm only.

3) L92~100: For the AOD algorithms in GK-2 sensors, the manuscript will introduce details of algorithm description including advantage and disadvantage. So, please reinforcement the purpose of this research.

4) Section 2: The manuscript explained the respective GK-2 sensors and applicable algorithm in the same sections. However, I suggest that the section will separate the instruments and algorithms.

5) L166: Please specify the uncertainty of AERONET AOD. "Uncertainty" means bias? Or precision?

6) Section 2.2: Most of AERONET sites are located on land surface. For the fusion, spatio-temporal homogeneity is important. However, the AERONET AOD, for the

reference dataset of training, is not spatially homogeneous. This inhomogeneity will affect the accuracy of fusion. In addition, please list-up or make a figure for AERONET sites.

7) Section 3.1: Why did the dataset re-gridding? Why don't you use the original pixel datasets for training?

8) L205: For cloud pixel identification, -28 K were used. Is that threshold also used in this study? As changing the instrument, the threshold value is also changed.

9) Section 4.1: I suggest that this section will move the method section.

10) Section 4.3.1: Why does the author separate the regions, AOD-EA and AOD-KO?

11) L390~L405: To write the AERONET site name, please write the location information.

12) Section 4.3.3: Author showed the diurnal variation of fusion AOD. However, this diurnal variation is not perfectly showed the diurnal variation. The data includes the arbitral signals during the fusion. How to be classified the real and arbitral diurnal variation from data?

13) Conclusion: Please add the further study of this research to improve the fusion results.

Minor comments

1) L15: "Geostationary Korea Multi-Purpose Satellite (GEO-KOMPSAT, GK)" → "second generation of Geostationary Korea Multi-Purpose Satellite (GEO-KOMPSAT-2, GK-2)"

2) Abstract (L22-L25): "The statistical and DNN-based ~~" is difficult to read. Please rephrase this sentence.

3) L35: Difference of definition between 'spectrometer' and 'radiometer' is different. However, in this manuscript, the author confused to use these words.

4) L134: In version 2 of the GEMS AOD at 550 nm, how to retrieve the AOD at 550 nm from GEMS? GEMS does not have 550 nm observation data.

5) L155: Please include the reference for GEMS algorithm.

6) L195: It is confused. Please re-describe.

7) L208: BTD10.3-13.3 is "BTD of the 10.3 and 13.3 um".

8) L209: "atmospheric window": 13.3 um is not an atmospheric window.

9) L288: "0.05+0.15AOD"

10) L303: Kim, M. et al. -> Kim et al.

11) L306: Please specify the wavelength of AOD. In addition, please clarify the wavelength of AOD in all description.

12) L349-L360: Please add the table of statistical AOD results before and after fusion.

13) L379: What is 'EE gradient'? Is that frequently used?

---

## Author Comment (AC1)

We would like to express appreciation to the reviewers for their insights and detailed review as well as for the suggested references. Our responses (in blue) for each comment (in black) are provided below.

Authors' response to RC1

The manuscript entitled by "AOD data fusion with Geostationary Korea Multi-Purpose Satellite (Geo-KOMPSAT) instruments GEMS, AMI, and GOCI-2: Statistical and deep neural network methods" are shown to the data fusion for various AOD retrieval products to enhance accuracy and stability of datasets. This manuscript showed details of statistical fusion methods and pre-processing before data fusion.

For the readability of manuscript, it needs to be checked the English correction. In addition, chapters of the manuscript must be re-constructed. In particular, some words and phrases are unclear. Although this manuscript is highly useful for the application of air quality and climate, the manuscript will be improved before publication. I summarized the details of comments.

Major comments

1) Title: This work used the AOD retrieval from GEMS, AMI, and GOCI-2, which are GK2 mission, not GK mission. I suggested that the title will change 'Geo-KOMPSAT- 2 (GK2)'. In addition, the author should check and correct this word in all manuscript.

We appreciate your suggestion; we specified the satellite mission as GK-2 on the title and also on the manuscript.

2) Introduction: This manuscript is purposed on showing the result of AOD data fusion. Although the manuscript explained several previous AOD retrieval algorithms, other product algorithms (such as aerosol index, SSA, ALH) are not essential to explain. Please simplify the introduction session more focusing on the AOD retrieval algorithm only.

Thank you for the insight. We agree that the data fusion studies outside of AOD are not essential. The introduction is simplified.

3) L92~100: For the AOD algorithms in GK-2 sensors, the manuscript will introduce details of algorithm description including advantage and disadvantage. So, please reinforcement the purpose of this research.

Thank you. Moved the corresponding paragraph in front of a paragraph regarding retrieval uncertainty due to observation wavelength.

4) Section 2: The manuscript explained the respective GK-2 sensors and applicable algorithm in the same sections. However, I suggest that the section will separate the instruments and algorithms.

We separated instrument section and algorithm description section as "2.1 GK-2 satellite instruments" and "2.2 Aerosol retrieval algorithm for GK-2 instruments". Also, details of the instrument are added.

5) L166: Please specify the uncertainty of AERONET AOD. "Uncertainty" means bias? Or precision?

Thank you. Specified as "the estimated uncertainty in precision" referring to Sinyuk et al. (2020).

6) Section 2.2: Most of AERONET sites are located on land surface. For the fusion, spatio-temporal homogeneity is important. However, the AERONET AOD, for the reference dataset of training, is not spatially homogeneous. This inhomogeneity will affect the accuracy of fusion. In addition, please list-up or make a figure for AERONET sites.

To assess whether the deep learning-based fusion accurately represents the spatial variation of AOD, we compared the fused AOD with a low Earth orbit satellite AOD product. We utilized reprocessed NOAA-20/VIIRS AOD data from the NOAA Environmental Data Record (EDR) system (Laszlo and Liu, 2022) for this comparison. VIIRS AOD data collected within 30 minutes of DNN AOD data were collocated for analysis. Figure RC1.1 illustrates that the DNN AOD aligns well with the VIIRS AOD, even for pixels that were not included in the training dataset due to sparse AERONET locations.

Also, AERONET site information is listed on Table 2.

[Figure]

**Figure RC1.1 Comparison of NOAA-20/VIIRS AOD at 550 nm and DNN AOD at 550 nm. Datasets for November 2022 are used, VIIRS AOD within 30 minutes from DNN AOD is collocated for comparison.**

7) Section 3.1: Why did the dataset re-gridding? Why don't you use the original pixel datasets for

training?

Because the geolocations of each instrument differ, re-gridding is essential to match them to a common geolocation grid. Additionally, we utilized re-gridded datasets for training to account for errors that may be induced during the re-gridding process.

We added explanations regarding the comment in Section 3.1.

8) L205: For cloud pixel identification, -28 K were used. Is that threshold also used in this study? As changing the instrument, the threshold value is also changed.

The -28 K threshold is set after testing cloud pixel identification with AMI level 1b data.

9) Section 4.1: I suggest that this section will move the method section.

Thank you for the suggestion. The section is moved to Section 3.1.

10) Section 4.3.1: Why does the author separate the regions, AOD-EA and AOD-KO?

Because of the limited field of regard of GOCI-II, the input AOD products of AOD-EA and AOD-KO are different, thus the resulting performance of fused products are different. Therefore, we conducted a separate analysis of the fused product.

11) L390~L405: To write the AERONET site name, please write the location information.

Done. Thank you.

12) Section 4.3.3: Author showed the diurnal variation of fusion AOD. However, this diurnal variation is not perfectly showed the diurnal variation. The data includes the arbitral signals during the fusion. How to be classified the real and arbitral diurnal variation from data?

The AERONET AOD on the day depicted in the manuscript reaches up to approximately 1.5, while the AOD at Yonsei_University and KORUS_UNIST_Ulsan exceeded 0.5 throughout the day. As indicated in the validation results of the fused AOD product, there is still an underestimation of high AOD values remaining after fusion. This discrepancy is attributable to the high uncertainty associated with input AOD at elevated levels. To mitigate this issue, improved input AOD data is necessary.

13) Conclusion: Please add the further study of this research to improve the fusion results.

Future study with more dataset and variables for DNN model is added as "The performance of aerosol data fusion can be improved with more dataset in the future study. For the MLE fusion, more sample leads to better representativeness of uncertainty weight for MLE. On the other hand, more dataset leads to better train performance of the DNN model. Moreover, DNN model in the future study will include more variables to predict optimal AOD."

Minor comments

1) L15: "Geostationary Korea Multi-Purpose Satellite (GEO-KOMPSAT, GK)" à"second generation of Geostationary Korea Multi-Purpose Satellite (GEO- KOMPSAT-2, GK-2)" 2) Abstract (L22-L25): "The statistical and DNN-based ~~" is difficult to read. Please rephrase this sentence.

Done. Thank you.

3) L35: Difference of definition between 'spectrometer' and 'radiometer' is different. However, in this manuscript, the author confused to use these words.

Changed to radiometer. Thank you.

4) L134: In version 2 of the GEMS AOD at 550 nm, how to retrieve the AMS not affected by the misclassification of the type?

In version 2 of the GEMS aerosol algorithm, AOD is retrieved at 443 nm and then extrapolated to 550 nm based on the aerosol optical properties of the selected type. Consequently, errors resulting from misclassification of aerosol type can impact the 550 nm AOD. However, the AOD error stemming from aerosol type misclassification is accounted for in the uncertainty estimation. Hence, it can be inferred that the fusion algorithm incorporates errors arising from type misclassification.

5) L155: Please include the reference for GEMS algorithm.

Reference for the GEMS algorithm is in the beginning of the paragraph as "The operational GEMS aerosol algorithm, based on real observations, was subsequently established by Cho et al. (2023)."

6) L195: It is confused. Please re-describe.

Rephrased as

"For temporal matching of 04:00 UTC fusion, GEMS AOD data scanned from 03:45 UTC to 04:15 UTC were utilized as the AOD representation for 04:00. In the case of AMI AOD, data were collected for a time span of 03:30−04:30 UTC from each precise hour and a median AOD was calculated. As for GOCI-II, data scanned at 03:15 and 04:15 were simply averaged."

7) L208: BTD10.3-13.3 is "BTD of the 10.3 and 13.3 um". 8) L209: "atmospheric window": 13.3 um is not an atmospheric window.

We appreciate pointing out confusing expression, thus rephrased as

"Detection of lower clouds involved the brightness-temperature difference (BTD) of the 13.3 and 10.3 μm (known as the "atmospheric window") bands (BTD10.3–13.3)."

9) L288: "0.05+0.15AOD" 10) L303: Kim, M. et al. -> Kim et al.

Done. Thank you.

There are two "Kim et al. 2021"s in the manuscript, two of them are identified as "Kim, M. et al." and "Kim, D. et al.".

11) L306: Please specify the wavelength of AOD. In addition, please clarify the wavelength of AOD in all description.

Specified as "Wavelength of AOD used for error analysis and data fusion are at 550 nm." In L XXX.

12) L349-L360: Please add the table of statistical AOD results before and after fusion.

Please refer to the Tables 5 and 6 added to the manuscript.

13) L379: What is 'EE gradient'? Is that frequently used?

The expression is fixed to "EE slope" and the value (0.15) is also specified.

---

## Author Comment (AC2)

We would like to express appreciation to the reviewers for their insights and detailed review as well as for the suggested references. Our responses (in blue) for each comment (in black) are provided below.

Authors' response to RC2

Overview

The paper compares AOD products from the GEMS, AMI, and GOCI-II instruments aboard the GEO-KOMPSAT and two fusion products of the single instrument retrievals with AOD AERONET observations and with MODIS DT data. The fusion products are optimised to match the AERONET observations using a deep neural network (DNN) or a maximum likelihood estimate (MLE). The error analysis is detailed and distinguishes between different AOD loads, NDVI, observation time and aerosol types. The authors find that the GOCI-II retrievals have the lowest error of the one-instrument retrievals and that the fused products using DNN has overall the smallest errors.

General remarks

The paper is very detailed and provides a lot of quantitative information about the error of the evaluated AOD products. But, the paper should provide more scientific information to help the reader to understand or interpret short-comings of the products or the choices for the fusion approach. For example, the choice of NDVI (and not other candidate parameters) as predictor of error needs to be discussed in more detail. Likewise, the determination of the aerosol type should be better explained.

Thanks for your comments. We understand that the choice of variables for uncertainty stratification is not appropriately explained. Thus, reasoning of the choices is added in the Section 3.2,

"As shown in Fig. 5-7, retrieval error does not increase (or decrease) linearly. Therefore, merging AOD datasets using the same RMSE value for all pixels is not desirable. The statistical fusion method linearizes the error characteristics by categorizing potential error sources such as AOD values, aerosol types, NDVI values, and observation times. The potential error source variables are selected based on previous studies with the following logistics. First, AOD value itself and aerosol type is selected because as aerosol loading increases, aerosol model assumption affects retrieval performance. Complex aerosol mixture at high aerosol loading leads to high uncertainty and aerosol retrieval algorithms have distinct aerosol model assumptions. NDVI is selected as possible error source to represent surface condition. Different surface types have different surface reflectance and surface types differentiate by vegetation amount and types (Hsu et al., 2013). Observation time difference in GEO measurements leads to distinct optical path of observed radiance. Therefore, GEO satellite AOD products have diurnal error variations (Lim et al., 2019; Zhang et al., 2020; Fu et al., 2023; Cho et al., 2024). To deal with the uncertainty from this, observation time is selected as the possible error source.".

The two fusion products have smaller error against AERONET observations than the single-instrument retrievals. This is perhaps not surprising because the fusion approaches were designed to match the AERONET observations and a prior bias correct of the single-instruments retrieval was performed. Such a correction procedure could also be applied to the individual satellite data sets. So, it remains unclear if the added benefit of the fusion approach is the AERONET-based error correction or the synergistic benefits of the MLE or DNN based methods to merge the products.

Fig. 1 and Fig. 2 are showing validation results of original AOD products for before and after bias correction, and for fused products. Both figures clearly show that the AERONET-based bias correction improves AOD. But after fusion of bias corrected AOD products, the quality of AOD improves even more. Also, tables for validation results are shown in Table 1 and Table 2. Simplified version of the tables is also added in the manuscript as "Table 5 and Table 6".

[Figure]

**Figure RC2.1 Validation results of AOD products within -EA region. Results of original AOD products before, after bias correction, and fused AOD products are shown. AOD products from November 2022 to April 2023 are used. The number of collocated points (N), linear regression equations, Pearson's correlation coefficient (R), root mean squared errors (RMSE), mean bias errors (MBE), and percentage within the expected error envelope (% within EE; EE: $\pm(0.05+0.15\ \tau_A)$) is shown. Dashed line and dotted lines indicate one-to-one line and expected error envelopes. Blue line indicates linear regression line of the satellite AOD and AERONET AOD.**

[Figure]

**Figure RC2.2 As for Fig. RC2.1, but for -KO region.**

**Table 1: Validation statistics of original AOD products and fused AOD products in -EA region. Validation period is from November 2022 to April 2023.**

|  | Before bias correction | | | Fused AOD | |
|---|---|---|---|---|---|
|  | GEMS | AMI-MRM | AMI-ESR | MLE AOD | DNN AOD |
| R | 0.800 | 0.834 | 0.860 | 0.888 | 0.905 |
| RMSE | 0.287 | 0.201 | 0.187 | -0.188 | 0.161 |
| MBE | -0.154 | -0.045 | -0.045 | -0.076 | -0.060 |
| % within EE | 39.5 | 52.1 | 53.3 | 60.6 | 65.6 |
|  | After bias correction | | | | |
| R | 0.838 | 0.846 | 0.870 | | |
| RMSE | 0.228 | 0.199 | 0.183 | | |
| MBE | -0.074 | -0.060 | -0.052 | | |
| % within EE | 51.3 | 54.1 | 55.8 | | |

**Table 2: Validation statistics of original AOD products and fused AOD products in -KO region.**

|  | Before bias correction | | | | Fused AOD | |
|---|---|---|---|---|---|---|
|  | GEMS | AMI-MRM | AMI-ESR | GOCI-II | MLE AOD | DNN AOD |
| R | 0.807 | 0.878 | 0.867 | 0.901 | 0.911 | 0.912 |
| RMSE | 0.187 | 0.129 | 0.129 | 0.114 | 0.113 | 0.102 |
| MLE | -0.086 | 0.017 | -0.002 | -0.038 | -0.047 | -0.028 |
| % within EE | 51.7 | 63.1 | 58.8 | 67.1 | 73.3 | 78.2 |

|  | After bias correction |  |  |  |  |
|---|---|---|---|---|---|
| R | 0.815 | 0.887 | 0.876 | 0.903 | |
| RMSE | 0.163 | 0.122 | 0.127 | 0.106 | |
| MLE | -0.025 | -0.025 | -0.009 | -0.011 | |
| % within EE | 57.5 | 66.2 | 63.4 | 75.5 | |

The paper uses a lot of acronyms for different versions of the retrievals and it is difficult for the reader to follow. For improved readability I suggest 1) to spell out more of the acronyms in the figure captions, 2) to add a table that that summarises the data sets and 3) to add a table that summarised the error measures (bias, RMSE etc) for all considered single or fused products to give a better overview     of the accuracy.

For 1), we revised figure captions. For 2), additional information of each aerosol products are added in Table 1. For 3), tables comparing the original AOD products and the fused AOD products are added as "Table 5 and Table 6". For the consistency of validation standard, the same level and period of AERONET dataset that is used for Fig. 8 is also used. Also, validation is done for two separate regions for -EA and -KO.

Specific comments:

L 140 Please provide more detail on the aerosol type classification. Why is GEMS not affected by the misclassification of the type?

Details of type classification of AMI and GOCI-II are further explained at the end of the Section 2.2.1 as "YAER algorithm first retrieves AODs at all wavelengths within UV-NIR range and converted to 550 nm for all aerosol types. Then, aerosol type that shows minimum variance at 550 nm are selected aerosol type for the corresponding inversion pixel.".

We understand the misleading description. All the aerosol retrieval algorithms are affected by misclassification. However, the wide wavelength coverage of AMI and GOCI-II are "more" sensitive to errors from aerosol size, while GEMS are "more" sensitive to errors from misclassification of absorbing/scattering aerosol types. So what we meant was that GEMS is less sensitive to error from misclassification of aerosol types that are in different size. Misleading expressions are revised in the manuscript.

L 159 How is the aerosol type derived?

Following explanation is added to the paragraph:

"Aerosol type is selected with UV aerosol index (UVAI) and VIS aerosol index (VISAI). The algorithm assigns NA type to pixels with low UVAI values. The other pixels are separated into highly absorbing fine (HAF) type and DU type according to the VISAI values."

L 164 Please comment on the differences and biases between AERONET version 3 level 2 and level 1.5

The comment on the differences on the AERONET data levels is added as

"The AERONET level 1.0 data are unscreened measurement data. The cloud and pointing error screening is applied to level 1.0 data to produce a level 1.5 dataset. The level 1.5 data series are raised to level 2.0 (quality-assured) series after final calibration values are applied and manual data inspection is completed."

L 178 Please motivate better the choice of NDVI. Fig 5, 6 and 7 (b) do not show a distinct relation between NDVI and error.

The motivation for the choice of NDVI and the other error sources are described in detail in the Section 3.2.

L 225 What is the procedure if an instrument product has no data?

As indicated on the table 3, fusion AOD is produced in accordance with the data availability.

L 226 All retrievals come from the same satellite. So, index i should represent the instrument or product.

Done. Thank you.

L 235 From which instrument was the aerosol type obtained?

Each aerosol product has their own aerosol types as side products.

L 245 This type classification should be explained earlier.

Type classification is explained in the Section 2.2.1 and 2.2.2.

L 265-280 It remains unclear what type of cloud masking was applied for the different products and if all problems related to cloud masking could be resolved. Please provide a summary in this section.

The paragraph is moved to Section 3.1 so that the readers can understand that the cloud masking procedures listed in Table 3 and that the cloud masking is applied to both GEMS and GOCI-II.

L 287 Please provide more detail on EE. What is its purpose? What is tau. Why does it make sense to use the MODIT DT approach here.

It is used as a common metric to evaluate multiple aerosol optical depth products at once. Also, the expected error envelope was calculated analytically by Levy et al. (2013). Authors clarified that the EE envelope is borrowed from the reference as "The expected error envelope (EE envelope ±(0.05 + 0.15AOD).) of AOD was established by Levy et al. (2013)"

Also, tau is replaced to "AOD".

L 343 Why were only the fused data processed and evaluated for the two different domains EA and KO? Which was the domain for the single instrument data?

The reason of the use of two different domains is explained as "The GOCI-II field of regard focusing on KO was smaller than those covering EA, so the fused AOD utilizing GOCI-II AOD was confined within the domain. Therefore, two groups of fused AOD products were generated: one involving the entire EA domain (AOD-EA), and the other focusing exclusively within KO (AOD-KO), which is the domain covered by GOCI-II.". The evaluation of original data at two domains are added in the Table 5 and Table 6.

L354 "The statistical fusion approach thus effectively accommodated nonlinearity in retrieval uncertainty, despite possibly not capturing all complexity in the data."   Please explain better what you mean. The fused data have the advantage of being optimized to match the AERONET data.

Revised as "By merging the original AOD dataset according to retrieval error compared to AERONET in different retrieval conditions (NDVI, observation time, aerosol loading and type), the statistical fusion approach thus effectively accommodated nonlinearity in retrieval uncertainty, despite possibly not capturing all complexity in the data.". Also, additional explanation is added in Section 3.2 "Statistical aerosol fusion: MLE AOD" as "As shown in Fig. 5-7, retrieval error does not increase (or decrease) linearly. Therefore, merging AOD datasets using the same RMSE value for all pixels is not desirable. The statistical fusion method linearizes the error characteristics by categorizing potential error sources such as AOD values, NDVI values, and observation times."

L 361-415 This section is perhaps to detailed and complicated. It would be sufficient to simply compare the AOD products and MODIS DT against AERONET and compare the errors for the different situations or locations.   It remains slightly unclear if the new fusion products have smaller errors than the MODIS DT retrievals over the study area.

We agree that the including MODIS DT to compare other products is confusing. Fig. 9 has been simplified to plot $1\sigma$ error vs. AOD. The overall description of section 4.2.2 is much simplified.

L 363 please explain the retrieval error. Is that the "theoretical" retrieval error provided by the retrieval algorithm or the error of the product against AERONET.     How is the theoretical retrieval error of the fused data set calculated.

Retrieval error is calculated as 68th percentile of AOD error of the product against AERONET. Description is added as "(68th percentile of absolute AOD error against AERONET, $|\Delta_S|_{68}$; 1σ of gaussian distribution)". Also, according to Sayer et al. (2020), the "retrieval error", which is 68th percentile, corresponds to $1\sigma$ of gaussian error distribution. Thus, the retrieval error can be regarded as theoretical retrieval error.

Figures:

Please include the statistical error measure of Fig 4 and Fig 8 in a table.

Table 5 and Table 6 are added in the manuscript.

It is not obvious what Fig 9 shows.

Fig. 9 is changed. Please refer to the response on L361-415.

Tables:

See general comment.

---

## Author Comment (AC3)

We would like to thank Ding Li for the comment as well as for the suggested papers. Our responses (in blue) for each comment (in black) are provided below.

**Authors' response to CC**

This document employs two distinct methods, Maximum Likelihood Estimation (MLE) and Deep Neural Networks (DNN), to integrate pixel-level uncertainties in the fusion of three Aerosol Optical Depth (AOD) products. The result is an improved hourly AOD dataset compared to individual AOD products, evident in its superior validation against ground-based AERONET AOD over East Asia. The approach meticulously addresses potential sources of errors and various challenges, positioning the resulting dataset to provide high-precision hourly Aerosol Optical Depth (AOD) products. Moreover, the article maintains a coherent logical flow and is enriched by visually appealing charts, rendering it an outstanding paper.

Major comments:

1. The amount of data is a barrier to machine learning models. Can we consider all types when there are not many AERONET sites in eastern Asia? It would be interesting to see a discussion on how the model performance varied with different volumes of data. Did the model performance improve with more data? If the data volume was a limitation in this study, it would be worth discussing how future work could overcome this. Are there plans to gather more data or use techniques like data augmentation?

   Thank you for the comment. We have done a preliminary deep learning model training with a 6-month dataset. Since the inputs for the deep learning model (each original AOD products) are highly correlated to the output of the training model, the result was comparable to the one in the manuscript. However, more data in the future may include a greater number of high AOD data to train, therefore can mitigate the remaining underestimation of DNN AOD. The future work regarding the data size is included in the conclusion.

2. For MLE (235 line): Based on this analysis, the bias of each AOD product was subtracted according to the NDVI value, selected aerosol type, and observation time. For DNN (245 line): This involved standardization of the NDVI, hour, and aerosol type index. Both models have taken into account the parameters mentioned above. However, in the figures presented in this document, there is an absence of accuracy analysis for diverse parameter combinations, with the focus solely on overall data analysis. Enhancing the article's significance can be achieved by incorporating analysis results for various parameter combinations and providing explanations for the observed outcomes.

   Thank you for the comment. Figure CC1 suggests the bias analysis of the two fused AOD products (MLE and DNN). We found out that the bias of the products was effectively mitigated after fusion. However, we thought the improved results are obvious, so did not include the figure to the manuscript.

[Figure]

**Figure CC1 AOD bias of MLE and DNN AOD relative to AERONET AOD, NDVI, selected aerosol type, and observation time. Whisker ends correspond to the 10th and 90th percentiles of the bin. Box ends correspond to the 25th and 75th percentiles. Horizontal lines in each box indicate bin median.**

**Numbers and bar plots in blue indicate the number of collocated AOD points in each box–whisker.**

---

## Referee Report (RR1)

This revised manuscript was mostly corrected the reviewers' comments, and the manuscript is well constructed. However, I would like to suggest that some additional corrections for the improvement of manuscript.

1) Section 2: The specifications of instruments are too omitted and not standardized. For Gk2B instruments, the manuscript wrote the wavelength, spatial coverage, and resolution. However, the AMI is not. Please add and unify the instrument explanations.

2) Section 2.2.2: Please add the advantage and disadvantage of type classification for GEMS AOD retrieval.

Minor Comments

1) L165: hyperspectral 'radiance' observations
2) L174: The GEMS → the GEMS
3) L191: AERONET AODs is known to be 0.010-0.021 → for version 3?
4) L198: Please add the reference

Park, S. S., Kim, S. -W., Song, C. -K., Park, J. -U., and Bae, K.: Spatio-Temporal Variability of Aerosol Optical Depth, Total Ozone and NO2 Over East Asia: Strategy for the Validation to the GEMS Scientific Products, Remote Sens, 2020, 12(14), 2256.

---

## Author Response (AR2)

We would like to express appreciation to the reviewers for their insights and detailed review as well as for the suggested references. Our responses (in blue) for each comment (in black) are provided below.

Authors' response to minor comments of referee #1

This revised manuscript was mostly corrected the reviewers' comments, and the manuscript is well constructed. However, I would like to suggest that some additional corrections for the improvement of manuscript.

1) Section 2: The specifications of instruments are too omitted and not standardized. For Gk2B instruments, the manuscript wrote the wavelength, spatial coverage, and resolution. However, the AMI is not. Please add and unify the instrument explanations.

The details of the AMI observations are added as "As a meteorological imager, AMI has spectral channels in the VIS–IR range (Kim et al., 2021), which is 3 VIS channels, 1 near-IR channel, and 10 IR channels from 0.47 μm to 13.3 μm. A 0.65 μm channel has 0.5 km spatial resolution, and 0.47, 0.51 μm channels has 1.0 km spatial resolution. The IR channels has 2.0 km spatial resolution. AMI scans full-disk every 10 minutes, and local area near Korean peninsula every 2 minutes.". (L112)

2) Section 2.2.2: Please add the advantage and disadvantage of type classification for GEMS AOD retrieval.

The advantage and disadvantage of the GEMS type classification is added as "The type classification of the GEMS AOD retrieval is superior to the other algorithms using visible wavelengths because of the sensitivity of UV wavelength to scattering characteristic of aerosols. Yet, relatively short range of observation wavelength in VIS region of GEMS compared to AMI and GOCI-II lacks sensitivity to aerosol size information.". (L180)

Minor Comments

1) L165: hyperspectral 'radiance' observations

Done. Thank you. (L168)

2) L174: The GEMS > the GEMS

Done. Thank you. (L177)

3) L191: AERONET AODs is known to be 0.010-0.021 à for version 3?

Yes, according to Giles et al. (2019) and Sinyuk et al. (2020), the AOD uncertainty was 0.01-0.02, depending on the wavelength. AOD uncertainty was spectrally dependent with the higher errors in the UV. We revised the manuscript to make it clear that the AOD uncertainty depends on the wavelength as "The estimated uncertainty in precision in AERONET AODs is known to be 0.010–0.021 depending on the wavelength". (L198)

4) L198: Please add the reference

Park, S. S., Kim, S. -W., Song, C. -K., Park, J. -U., and Bae, K.: Spatio-Temporal Variability of Aerosol Optical Depth, Total Ozone and NO2 Over East Asia: Strategy for the Validation to the GEMS Scientific Products, Remote Sens, 2020, 12(14), 2256

Done. Thank you. (L203)